# Turbulent kinetic energy dissipation rate and associated fluxes in the western tropical Atlantic estimated from ocean glider observations

Peter M.F. Sheehan[1], Gillian M. Damerell[1], Philip J. Leadbitter[1], Karen J. Heywood[1], and Rob A. Hall[1]

[1]Centre for Ocean and Atmospheric Sciences, School of Environmental Science, University of East Anglia, Norwich, NR4 7TJ, United Kingdom

**Correspondence:** Peter Sheehan (p.sheehan@uea.ac.uk)

**Abstract.** Ocean gliders enable us to collect the high-resolution microstructure observations necessary to calculate the dissipation rate of turbulent kinetic energy, $\varepsilon$, on timescales of weeks to months: far longer than is normally possible using traditional ship-based platforms. Slocum gliders have previously been used to this end; here, we report the first detailed estimates of $\varepsilon$ calculated using the Batchelor spectrum method on observations collected by a FP07 fast thermistor mounted on a Seaglider. We use these same fast thermistor observations to calculate $\varepsilon$ following the Thorpe scale method and find very good agreement between the two methods. The Thorpe scale method yields larger values of $\varepsilon$, but the average difference, which is less than an order of magnitude, is smaller than reported elsewhere. The spatio-temporal distribution of $\varepsilon$ is comparable for both methods. Maximum values of $\varepsilon$ ($10^{-7}$ W kg$^{-1}$) are observed in the surface mixed layer; values of approximately $10^{-9}$ W kg$^{-1}$ are observed between approximately 200 and 500 m depth. These two layers are separated by a 100 m thick layer of low $\varepsilon$ ($10^{-10}$ W kg$^{-1}$), which is co-located with a high-salinity layer of Subtropical Underwater and a peak in the strength of stratification. We calculate the turbulent heat and salt fluxes associated with the observed turbulence. Between 200 and 500 m, $\varepsilon$ induces downward fluxes of both properties that, if typical of the annual average, would have a very small influence on the heat and salt content of the overlying salinity-maximum layer. We compare these turbulent fluxes with two estimates of double-diffusive fluxes that occur in regions susceptible to salt fingers, such as the western tropical Atlantic. We find that the double-diffusive fluxes of both heat and salt are larger than the corresponding turbulent fluxes.

## 1 Introduction

Turbulence in the ocean, and the mixing of different water masses that it induces, are of fundamental importance to ocean dynamics. Over relatively small scales, turbulent mixing often controls the distribution of key water mass properties and tracers; over the world ocean, the sum of these small-scale processes is responsible for the closure of the thermohaline circulation and for the primary production that relies on the upward flux of nutrients to the euphotic zone.

Estimating the dissipation rate of turbulent kinetic energy, $\varepsilon$, from high-resolution observations of shear and temperature (e.g. Lueck et al., 2002; Oakey, 1982; Ruddick et al., 2000) has, historically, required considerable ship time, plus specialist instruments and expertise, so there are relatively few of such estimates. Methods such as Thorpe scaling (Thorpe, 1977) and finescale parameterisation (Polzin et al., 2014; Whalen et al., 2015) have been developed to enable $\varepsilon$ to be estimated from

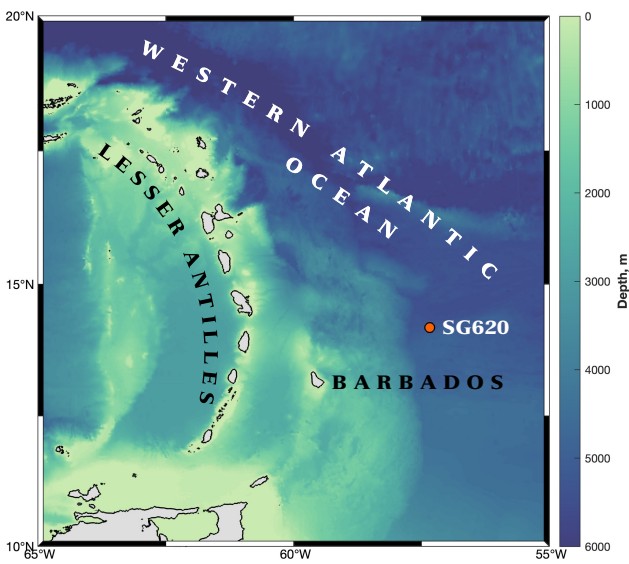

**Figure 1.** Bathymetry (m) of the western tropical Atlantic and the the eastern Caribbean Sea in the region of the Lesser Antilles. The location of SG620, northeast of Barbados, is marked by the orange circle. Land is shaded grey.

standard CTD and ADCP observations of temperature, salinity and current velocity – hereafter referred to as finescale methods and observations. Although finescale methods do not require specialist instruments (e.g. Fer et al., 2010b; Whalen et al., 2012, 2015), they are dependent on more assumptions; their results tend not to be valid over as wide a range of conditions as those derived from high-resolution, microstructure-scale observations (Polzin et al., 2014; Whalen, 2021) – hereafter referred to as microstructure methods and observations. Thus, despite the widespread use of finescale methods, the potential remains for discrepancies between estimates of $\varepsilon$ based on finescale and microstructure observations (Howatt et al., 2021).

Given the proliferation in the use of buoyancy-driven ocean gliders over the last decade, there is growing interest in using them to collect microstructure observations. Because of a glider's smooth flight through the water column, it resembles the free-falling, loosely tethered profilers traditionally used to collect microstructure observations. A growing body of literature makes use of microstructure observations collected by gliders, as well as setting out the best ways of processing such data sets (e.g. Fer et al., 2010b; Peterson and Fer, 2014; Palmer et al., 2015; Schultze et al., 2017; Scheifele et al., 2018; Scott et al., 2021). Up until now, the majority of studies have used microstructure shear observations collected by Slocum gliders (Palmer et al., 2015). Some authors have used microstructure temperature observations to estimate $\varepsilon$ (e.g. Peterson and Fer, 2014; Scheifele et al., 2018), while Rainville et al. (2017) briefly discuss the microstructure system developed for use on Seagliders, another commonly used glider platform, and present estimates of the rate of dissipation of temperature variance, $\chi$. However, observational studies of turbulence using autonomous platforms other than Slocum gliders are known to be lacking (Frajka-Williams et al., 2021). Here, we report in detail the first microstructure-based estimates of $\varepsilon$ calculated from Seaglider

microstructure temperature observations, and we compare the results with estimates of $\varepsilon$ calculated by applying the Thorpe scale method, commonly applied to finescale temperature and salinity observations, to the same microstructure observations.

The western tropical Atlantic (Fig. 1) is known for the persistent presence of the salt fingering regime of double-diffusion (Schmitt et al., 1987; Rollo et al., 2022). For salt fingering to occur, warm, saline water must overlie cooler, fresher water: the water column is therefore stably stratified by temperature but unstably stratified by salinity. Such conditions are maintained in the western tropical Atlantic by the presence of Subtropical Underwater (SUW) at the base of the mixed layer, a warm, high-salinity water mass common to tropical regions (Schmitt et al., 1987; Fer et al., 2010a). Beneath SUW, temperature and salinity both decrease with depth. In a salt fingering regime, the slow molecular diffusion of salt relative to the fast diffusion of heat leads to the development of salt fingers: narrow, small-scale filaments of alternately upwelling warming water and downwelling cooling water. Over time, double-diffusive convection and salt fingers promote the formation of thermohaline staircases: temperature and salinity profiles characterised by a series of homogeneous mixed layers separated by sharp, narrow gradient layers. Such staircases have previously been observed in the western tropical Atlantic (Schmitt et al., 1987; Rollo et al., 2022). Importantly for studies of ocean mixing, double diffusive convection enables the vertical transport of heat and salt by a mechanism other than the mechanical, turbulent mixing captured by $\varepsilon$.

Here, we use microstructure temperature observations collected by a Seaglider to estimate $\varepsilon$ using the Batchelor spectrum method (Sec. 2.2; Batchelor, 1959) and using the Thorpe scale method (Sec. 2.3; Thorpe, 1977), and compare the results (Sec. 3.1). From these estimates of $\varepsilon$, we derive turbulent fluxes of heat and salt through an observed layer of elevated $\varepsilon$ (Sec. 3.2), and compare these with heat and salt fluxes driven by salt fingers and double-diffusive mixing (Sec. 3.3). We discuss the results in Sec. 4.

## 2 Data and methods

### 2.1 Glider observations

As part of the EUREC4A field campaign (Stevens et al., 2021), Seaglider 620 was deployed at 14.2°N, 57.3°W, approximately 200 km northeast of Barbados (Fig. 1) on 23 January 2020. It completed 131 dives to 750 m before being recovered on 5 February 2020. The glider carried an unpumped CT sail measuring in situ conductivity and temperature, and a microstructure system. Given the shape of the Seaglider's hull, it is not possible to mount an all-in-one microstructure payload, such as the Rockland Scientific International (RSI) MicroRider that is used on Slocum gliders (e.g., Fer et al., 2014; Schultze et al., 2017; Scheifele et al., 2018). Instead, a reconfigured payload is used, one consisting of a pair of RSI MicroPod sensor modules mounted either side of the CT sail, and a dedicated pressure housing containing the system's DataLogger mounted inside the Seaglider's aft fairing (Creed et al., 2015). The system draws its power from the Seaglider and can be turned on and off on a dive-by-dive basis.

During the EUREC4A campaign, the glider was equipped with one MicroPod carrying a shear probe and one MicroPod carrying an FP07 fast-response temperature sensor. There was a fault with the shear probe on this deployment and the observations could not be used. The fast thermistor sensor samples at 512 Hz and has a sensitivity of better than 0.1 mK (Sommer et al.,

2013); thermal inertia of the sensor is such that its effective resolution is estimated to be 10 ms (i.e. 100 Hz; Sommer et al., 2013). Microstructure temperature observations are better suited than shear observations to estimating $\varepsilon$ in low-dissipation environments (Scheifele et al., 2018), and have the added advantage of being less readily contaminated by platform vibration (Frajka-Williams et al., 2021); here we focus on the temperature-based estimates of epsilon. The glider's hydrodynamic flight model, which is used to estimate along-path speed, is tuned following Frajka-Williams et al. (2011), and the thermal lag of the

standard CT sail is corrected following Garau et al. (2011).

## 2.2    Estimating $\varepsilon$ using the Batchelor spectrum method

We estimate $\varepsilon$ from the glider's fast thermistor temperature microstructure observations using the Batchelor spectrum method; we hereafter refer to these estimates as $\varepsilon_\mu$. For this, we use the Matlab toolbox produced by Benjamin Scheifele and Jeffrey Carpenter (github.com/bscheife/turbulence_temperature) and recently used by Howatt et al. (2021). The method is described

in detail by Scheifele et al. (2018), and much of the underlying theory, and a similar methodology, are described by Peterson and Fer (2014), so we here give only an outline.

We divide the temperature time series from the fast thermistor into half-overlapping segments of 32 seconds length. Within each 32-second segment, we further divide the measurements into 15 four-second, half-overlapping sub-segments. From each sub-segment, we calculate a temperature power spectrum, $\Delta_4$. We then average these 15 $\Delta_4$ to produce one power spectrum,

$\Delta_{32}$, that is representative of the original 32-second segment. Values of $\Delta_{32}$ at high frequencies, where the thermal inertia of the fast thermistor is such that its temporal response is inadequate, are corrected using the transfer function of Sommer et al. (2013). We convert each $\Delta_{32}$ from frequency space to wavenumber space (Fig. 2) using the glider's along-path speed averaged over the same 32 seconds, and assuming the validity of Taylor's frozen turbulence hypothesis (Scheifele et al., 2018).

We transform each $\Delta_{32}$ into a temperature-gradient spectrum, $\Psi = (2\pi k)^2 \Delta_{32}$, which should resemble the Batchelor spec-

trum, $\Psi_B$ (Batchelor, 1959), the theoretical spectrum that describes temperature-gradient spectra and which is commonly used when calculating $\varepsilon_\mu$ (e.g. Oakey, 1982; Ruddick et al., 2000; Peterson and Fer, 2014; Scheifele et al., 2018). The Batchelor spectrum is a function of $k_B$, the Batchelor wavenumber, and of $\chi$, the rate of destruction of temperature variance (Osborn and Cox, 1972). A comprehensive mathematical treatment of the use of $\Psi_B$ when estimating $\varepsilon$ is given by Peterson and Fer (2014). Here, we require $k_B$ in order to calculate $\varepsilon_\mu$ (W kg$^{-1}$) according to:

$$\varepsilon_\mu = \nu D_T^2 (2\pi k_B)^4 \tag{1}$$

where $\nu$ is the kinematic viscosity of seawater, $D_T = 1.44 \times 10^{-7}$ m$^2$ s$^{-1}$ is the molecular diffusion coefficient of temperature. We calculate $\chi$ according to:

$$\chi = \chi_l + \chi_{obs} + \chi_u$$
$$= 6 D_T \left( \int_0^{k_l} \Psi_B \, dk + \int_{k_l}^{k_u} \Psi \, dk + \int_{k_u}^{\infty} \Psi_B \, dk \right) \tag{2}$$

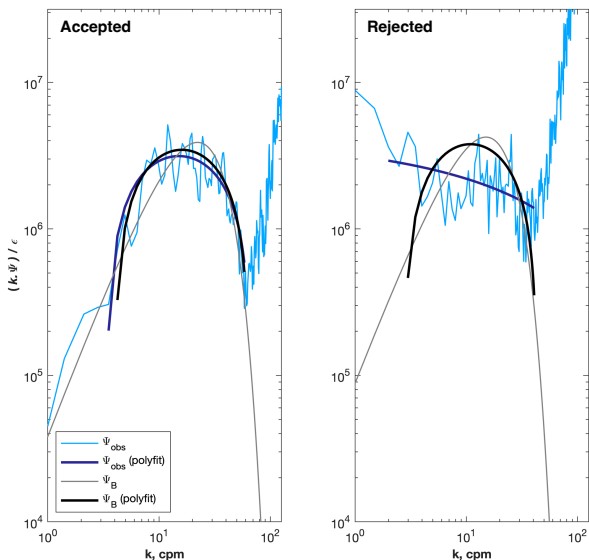

**Figure 2.** Example of temperature spectra, $\Psi$, that were accepted (left) and rejected (right) by the quality control algorithm. Observed and theoretical spectra are shown by the thin, light-coloured lines; the second-order polynomial fits are shown by thick, dark-coloured lines.

where $\chi_{obs}$ is that part of $\chi$ derived by integrating $\Psi$, and $\chi_l$ and $\chi_u$ are correction terms derived from $\Psi_B$. The factor of six comes from assuming isotropic turbulence. The wavenumbers $k_l$ and $k_u$ are, respectively, the lower and upper wavenumber limits of the range over which $\Psi$ is considered reliable; the criteria for choosing $k_l$ and $k_u$ are explained fully by Scheifele et al. (2018). Given an estimate of $\chi$, the maximum likelihood estimation procedure of Ruddick et al. (2000) is used to find the value of $k_B$ corresponding to the $\Psi_B$ that is the best fit to $\Psi$ between $k_l$ and $k_u$. On the first iteration, $\chi_l$ and $\chi_u$ are set to zero and hence $\chi = \chi_{obs}$. On subsequent iterations, the previous value of $\chi$ and the previous best-fit value of $k_B$ are used to estimate $\Psi_B$ and hence $\chi_l$ and $\chi_u$, and the estimate of $k_B$ is further refined.

An observed spectrum that deviates from the shape of the relevant theoretical spectrum should not be used to estimate $\varepsilon_\mu$. To discriminate between acceptably and unacceptably shaped spectra, we fit second order polynomials of the form:

$$P(k) = a.log_{10}(k)^2 + b.log_{10}(k) + c \tag{3}$$

to both the observed and the theoretical spectra, following the method of Scott et al. (2021), where $k$ is wavenumber and $a$, $b$ and $c$ are the polynomial coefficients to be determined. Prior to fitting, we normalise each spectrum by dividing by its corresponding estimate of $\varepsilon_\mu$; this enables the same criteria to be used when assessing goodness-of-fit over spectra that otherwise span many orders of magnitude. We also multiply spectra by $k$ in order to preserve variance. $P$ is defined over the same range of wavenumbers over which the observed spectrum is integrated when estimating $\chi$.

We accept a spectrum if:

1. The value of $a$ fitted to $\Psi$ is positive. (Note that $a$ fitted to $\Psi_B$ is always positive.)

2. The ratio of the $a$ values fitted to $\Psi_B$ and $\Psi$ ($a_{\Psi_B}/a_{\Psi}$) is less than two.

In addition, following Scheifele et al. (2018), we remove an estimate of $\varepsilon_\mu$ if:

3. Fewer than six points are included in the spectra fit.

4. If the quantity $U/(\varepsilon_\mu/N)^{1/2}$ is less than five, where $U$ is the glider's speed-in-direction-of-travel and $(\varepsilon_\mu/N)^{1/2}$ is an estimate of the turbulent flow velocities (Fer et al., 2014), indicating that Taylor's frozen turbulence hypothesis is invalid.

5. If the sum of the correction terms $\chi_u$ and $\chi_l$ is greater than the observed term $\chi_{obs}$ (Eqn 2).

Finally, following Peterson and Fer (2014), we remove an estimate of $\varepsilon_\mu$ if:

6. The mean absolute deviation, which quantifies the goodness of fit between $\Psi_{obs}$ and $\Psi_B$, is greater than $2(2/d)^{1/2}$, where $d$ is the degrees of freedom, calculated as 1.9 multiplied by the number of sub-segments within each 32-second segment, i.e., $1.9 \times 15$.

7. The estimate of $\varepsilon_\mu$ is greater than $2 \times 10^{-7}$ W kg$^{-1}$; even having corrected estimates of $\Delta_{32}$ at high frequencies, the effective resolution of the fast thermistor (100 Hz) is such that values of $\varepsilon_\mu$ greater than this cannot be reliably estimated.

Examples of accepted and rejected spectra are presented in Fig. 2. After quality control, 84% of $\varepsilon_\mu$ estimates remained. Quality-controlled estimates of $\varepsilon_\mu$ were binned, profile by profile, into 25 m vertical bins; we use the geometric mean in preference to the arithmetic (i.e. ordinarily used) mean, the better to represent the average of observations that span many orders of magnitude and which are not normally distributed.

## 2.3 Thorpe scale estimates

We apply the Thorpe scale method (Thorpe, 1977) to the fast thermistor microstructure potential temperature observations to derive a second, independent estimate of the turbulent kinetic energy dissipation rate, hereafter referred to as $\varepsilon_T$. Given that the effective resolution of the fast thermistor is estimated to be 100 Hz, we apply a low-pass, 12th-order Butterworth filter with a cut-off frequency of 100 Hz to remove the highest-frequency variability. This prevents instrumental noise erroneously manifesting as small density overturns (Mater et al., 2015; Ijichi and Hibiya, 2018). Temperature observations were then binned into 10 ms bins using the arithmetic mean, giving an effective vertical resolution of $3 \pm 0.5$ mm.

Each temperature profile is then re-ordered in depth so that temperature always decreases with depth (i.e. is stable with respect to temperature). From the re-ordered profile we calculate the vertical Thorpe displacement, $\Delta z$: the difference between an observation's original depth and its re-ordered depth. We identify an overturn as a vertical segment in which the cumulative sum of $\Delta z$ is non-zero, and which is bounded above and below by segments in which the cumulative sum of $\Delta z$ is zero. Following Ijichi and Hibiya (2018) and Howatt et al. (2021), we combine all overturns that are smaller than 2 m and are within

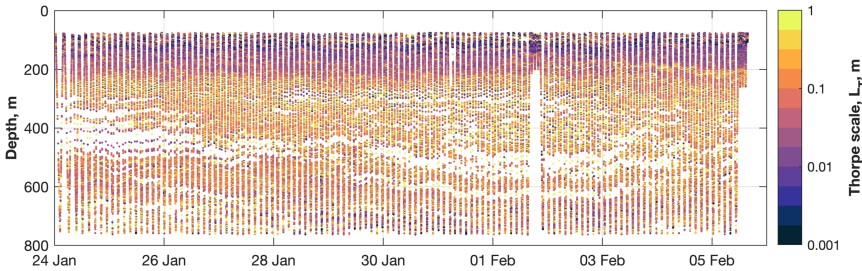

**Figure 3.** Thorpe scale estimates, $L_T$ (m). Higher values of $L_T$ are indicative of higher dissipation, $\varepsilon_T$.

1 m of an adjacent overturn until the region is larger than 2 m. Overturns smaller than 2 m that are further than 1 m from an adjacent overturn are not aggregated. This assumes that a succession of small overturns is the signal of a once larger, single overturn that has recently decayed into a series of smaller overturns (Smyth et al., 2001). The Thorpe scale, $L_T$ (Fig. 3), is then the root mean square of $\Delta z$ over an overturn:

$$L_T = \langle \Delta z^2 \rangle^{1/2} \tag{4}$$

where angular brackets indicate the mean. Larger values of $L_T$ correspond to larger mixing rates; the largest values of $L_T$ (approximately 1 m) are found principally between 200 and 500 m, although much lower values (approximately 0.01 m and lower) are also found within this depth range (Fig. 3). Similarly high values of $L_T$ are also found in the bottom 200 m sampled towards the beginning of the deployment. Above 200 m, values of $L_T$ are much lower, being generally below 0.01 m (Fig. 3).

Finally, we calculate $\varepsilon_T$ (W kg$^{-1}$) from $L_T$ by relating the Ozmidov scale, $L_O = (\varepsilon_T/N^3)^{1/2}$ (Ozmidov, 1965), to $L_T$ by
the empirical relation $L_O = 0.8\, L_T$ (Dillon, 1982), which yields:

$$\varepsilon_T = 0.64 L_T^2 N^3 \tag{5}$$

where $N$ is the background buoyancy frequency calculated using the Seaglider's finescale temperature and salinity observations, binned into 5 m vertical bins, then smoothed in the vertical using a Gaussian-windowed running mean over nine bins (i.e. 45 m).

Our estimates of $\varepsilon_T$ are derived from potential temperature rather than potential density; we must therefore assume that temperature is the dominant control on density. In regions where this is not the case – i.e. in regions where salinity is the dominant control on density – temperature perturbations may not correspond to the density perturbations that the Thorpe scale method takes to be indicative of turbulent overturns. To identify regions where salinity is the dominant control on density, we use the density ratio, $R_\rho$:

$$R_\rho = \frac{\alpha \Theta_z}{\beta S_z} \tag{6}$$

where $\alpha$ is the thermal expansion co-efficient, $\Theta_z$ is the vertical temperature gradient, $\beta$ is the haline contraction coefficient and $S_z$ is the vertical salinity gradient. We calculate $\Theta_z$ and $S_z$ from finescale glider observations binned into 5 m vertical bins using the arithmetic mean, and find $R_\rho$ at the depth of each overturn. Where $-1 < R_\rho < 1$, salinity is the dominant control on density, and we discard any overturns and associated value of $\varepsilon_T$. In total, 2388 overturns are discarded by the $R_\rho$ quality control criterion, 4.03% of the total. In the majority of the water column, temperature is the dominant control on density, and so temperature observations may be reliably used to estimate $\varepsilon$. Estimates of $\varepsilon_T$ that are discarded correspond principally to large values of $L_T$ ($> 1$ m) in mid-depth regions (i.e. between 200 and 600 m). This is the part of the water column in which the majority of the thermohaline staircases are found (Rollo et al., 2022).

Finally, we discard all values of $\varepsilon_T$ shallower than 75 m because a temperature inversion in the mixed layer is erroneously identified as an overturn. Remaining estimates of $\varepsilon_T$ are binned into 25 m vertical bins using the geometric mean, as for $\varepsilon_\mu$

## 3   Results

### 3.1   Estimates of $\varepsilon$

The water masses observed are typical of the region (e.g. Schmitt et al., 1987). A warm ($> 26$ °C) surface mixed layer of intermediate salinity overlies SUW, a salinity-maximum ($> 37.6$ g kg$^{-1}$) layer located in the upper thermocline (Fig. 4). Beneath SUW, temperature and salinity steadily decrease with depth into the Antarctic Intermediate Water layer that lies beneath (Fig. 4). Two maxima in buoyancy frequency are observed: an upper maximum at the base of the surface isohaline layer, and a lower maximum at the base of the surface isothermal layer (Fig. 4).

There is generally good agreement between $\varepsilon_\mu$ and $\varepsilon_T$. Histograms of the two distributions are very similar and, when scattered one against the other, the points are clustered around the one-to-one line (Fig. 6a and b). The principal difference is at depths between 400 and 600 m in the first two days of the deployment, when $\varepsilon_\mu$ ($> 10^{-9}$ W kg$^{-1}$) is noticeably higher than $\varepsilon_T$ ($< 10^{-10}$ W kg$^{-1}$; Fig. 5). This is a region with a clear pattern in the differences between $\varepsilon_\mu$ and $\varepsilon_T$: in many other regions, the differences are fairly randomly distributed (Fig 5c).

Of the two estimates of $\varepsilon$, the higher of the two is $\varepsilon_T$, the geometric mean of which is $3.42 \times 10^{-9}$ W kg$^{-1}$; the geometric mean of $\varepsilon_\mu$ is $3.05 \times 10^{-9}$ W kg$^{-1}$. We note that, on average, the difference between the two estimates is small, and much less than an order of magnitude. The more variable of the two is $\varepsilon_\mu$, the geometric standard deviation factor (GSDF) of which is 5.58. The GSDF of $\varepsilon_T$ is 4.43. (Note that GSDF is multiplicative, not additive, and is therefore dimensionless. The range is from the *geometric mean/GSDF* to *geometric mean* $\times$ *GSDF*.) Averages and GSDFs are calculated from bins only where estimates of both $\varepsilon_\mu$ and $\varepsilon_T$ are available.

The highest values of $\varepsilon$ are found in approximately the top 50 m of the water column, in the surface mixed layer above the upper boundary of SUW ($> 10^{-8}$; Fig. 5; $\varepsilon_\mu$ only). The upper boundary of SUW corresponds to the shallowest band of high buoyancy frequency (Fig. 4); a peak in the strength of the stratification might be expected to arrest the downward penetration of surface mixing. But these highest values are infrequently observed: a large proportion of $\varepsilon_\mu$ estimates in the mixed layer are greater than $2 \times 10^{-7}$, beyond the range for which the FP07 fast thermistor observations and the Batchelor spectrum method

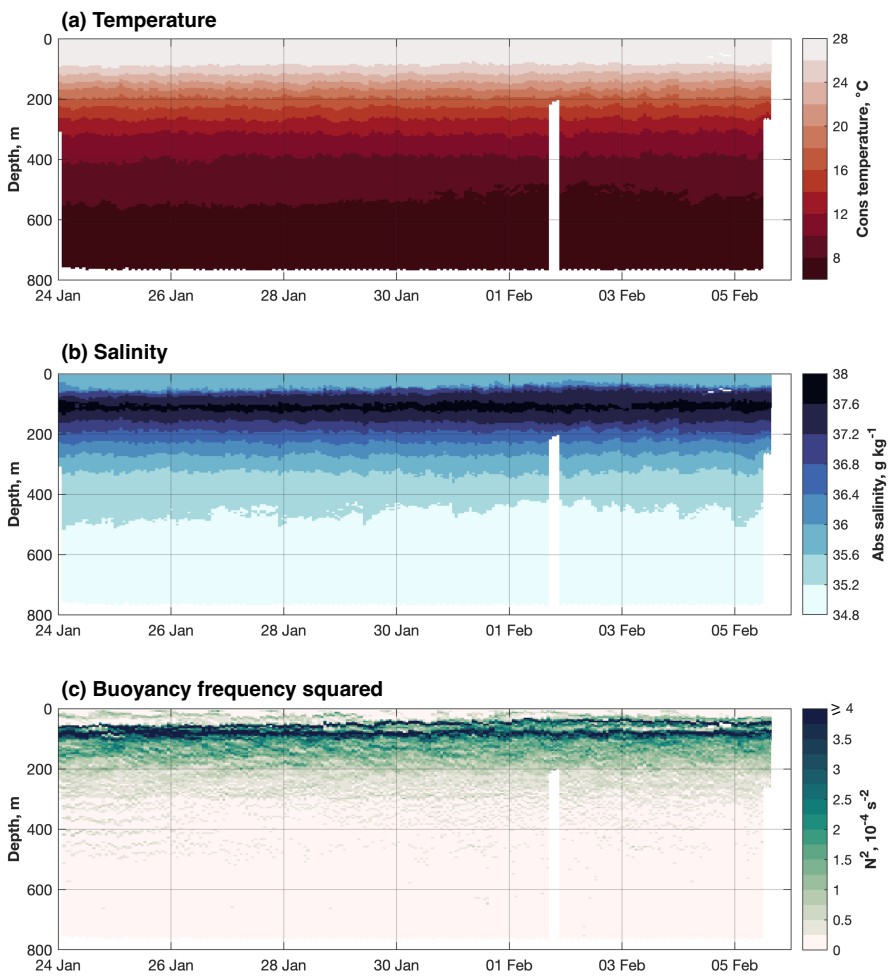

**Figure 4.** Time series of hydrographic observations from SG620, averaged (arithmetic mean) into 5 m bins: **(a)** conservative temperature (°C), **(b)** absolute salinity (g kg$^{-1}$), and **(c)** buoyancy frequency squared ($N^2$; s$^{-2}$).

yield meaningful results (Sec. 2.2; Peterson and Fer, 2014). Consequently, the remaining values are all below this threshold; indeed, considerable variability in $\varepsilon_\mu$ is observed in the mixed layer – this is reflected in the relatively high GSDF for $\varepsilon_\mu$ in this region (Fig. 5) – and many remaining estimates are low ($< 10^{-10}$). The mean $\varepsilon_\mu$ in the upper water column is therefore likely to be biased towards these low values. In the remainder of the water column, $\varepsilon_\mu$ and $\varepsilon_T$ are predominantly below $2\times 10^{-7}$ (Fig. 5), hence the comparison between the two is meaningful.

Between approximately 100 and 200 m, a thin layer with moderate values of $\varepsilon$ lies within SUW in the upper thermocline (Figs. 4 and 5). Here, values of both $\varepsilon_\mu$ and $\varepsilon_T$ are commonly between $10^{-9.5}$ and $10^{-9}$ W kg$^{-1}$. Below this low-$\varepsilon$ SUW layer, between approximately 200 and 500 m, is a relatively thick layer with higher values of $\varepsilon$ ($10^{-9} < \varepsilon < 10^{-8}$ W kg$^{-1}$; Fig. 5).

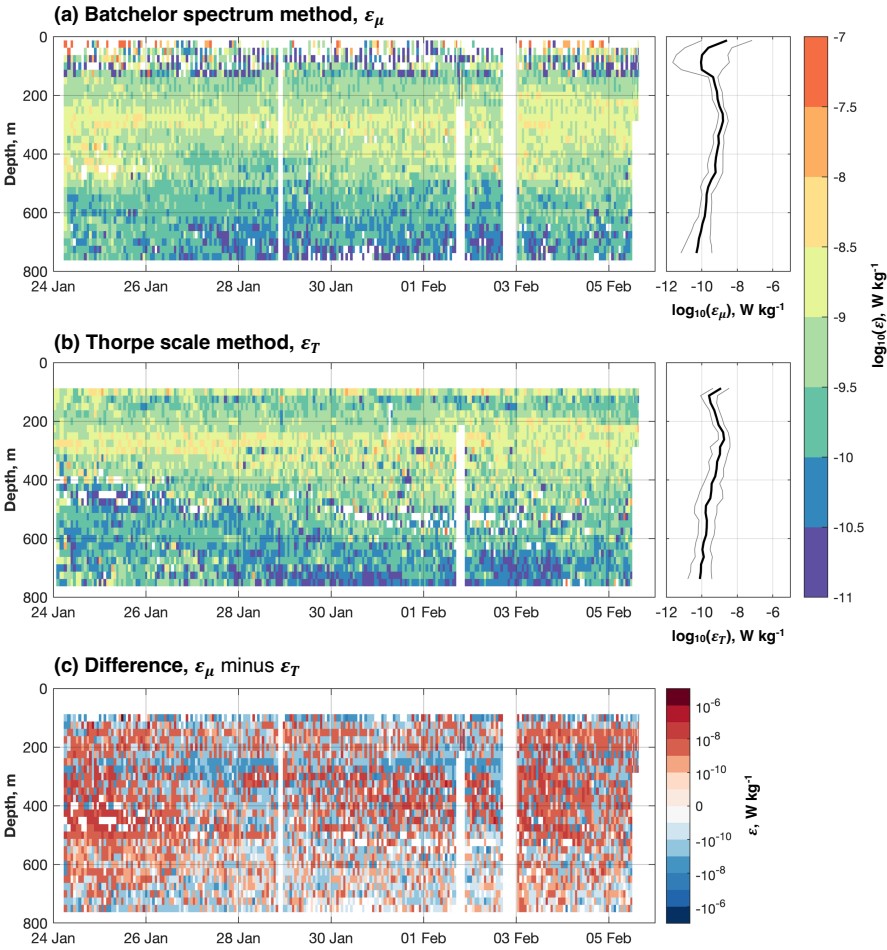

**Figure 5.** Turbulent kinetic energy dissipation rate, $\varepsilon$ (W kg$^{-1}$) as estimated using **(a)** the Batchelor spectrum method ($\varepsilon_\mu$) and **(b)** the Thorpe scale method ($\varepsilon_T$). The respective geometric means (thick lines) and geometric standard deviation factors (thin lines) are shown in the panels on the right. Note that geometric standard deviation factor is multiplicative. **(c)** The difference between $\varepsilon_\mu$ and $\varepsilon_T$.

Values of $\varepsilon_T$ in this layer are generally higher than values of $\varepsilon_\mu$ (Fig. 5), which would explain why the distribution of $\varepsilon_T$ is slightly skewed to higher values than the distribution of $\varepsilon_\mu$ (Fig. 6a). A few values of $\varepsilon_T$ are in excess of $10^{-7.5}$ W kg$^{-1}$ (Fig. 5). The thickness of this higher-$\varepsilon_\mu$ and -$\varepsilon_T$ layer increases by 50 to 100 m over the course of the deployment. Below 700 m, both

$\varepsilon_\mu$ and $\varepsilon_T$ are less than $10^{-10}$ W kg$^{-1}$ between 28 January and 4 February; the differences between the two estimates also tend to be lower within this spatio-temporal range (Fig. 5c). This is in contrast to higher values of $\varepsilon_\mu$ and $\varepsilon_T$ ($> 10^{-10}$ W kg$^{-1}$) within the 700 to 800 m depth range at the beginning and end of the deployment.

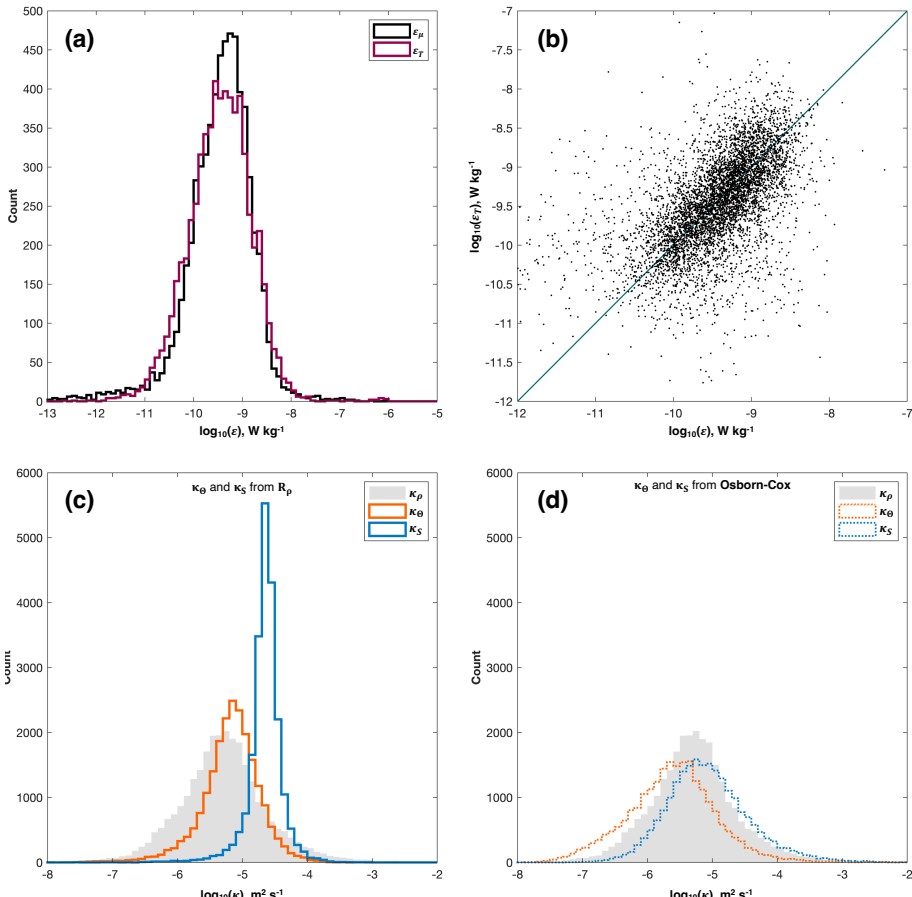

**Figure 6. (a)** Histograms of turbulent kinetic energy dissipation rate as estimated using the Batchelor spectrum method ($\varepsilon_\mu$; green line) and the Thorpe scale method ($\varepsilon_T$; purple line; both W kg$^{-1}$). **(b)** Scatter plot of $\varepsilon_T$ against $\varepsilon_\mu$. The one-to-one line is shown in dark green. **(c)** Histograms of thermal diffusivity ($\kappa_\Theta$; orange line) and haline diffusivity ($\kappa_S$; blue line) as estimated from the density ratio, $R_\rho$. Diffusivity, $\kappa_\rho$, estimated from the Osborn relation and $\varepsilon_\mu$ is plotted as the shaded grey histogram. **(d)** Histograms of $\kappa_\Theta$ (orange dotted line) and $\kappa_S$ (blue dotted line) as estimated from $\chi$ using the Osborn-Cox relation. Diffusivity estimated from the Osborn relation is plotted as the shaded grey histogram, as in (c).

## 3.2 Heat and salt fluxes

We calculate diapycnal diffusivity, $\kappa_\rho$, from $\varepsilon$ using the Osborn relation:

$$\kappa_\rho = \Gamma \frac{\varepsilon}{N^2} \tag{7}$$

where $\Gamma$ is mixing efficiency, which is here taken to be 0.2 (Osborn, 1980). We use $\varepsilon_\mu$ in preference to $\varepsilon_T$ because the former has better coverage in the mixed layer (Fig. 5).

Below approximately 250 m, the distribution of $\kappa_\rho$ resembles that of $\varepsilon_\mu$ due to the relatively low variability in $N^2$ at these depths (Fig. 4c). Above 200 m, $N^2$ increases substantially in the pycnocline and thus $\kappa_\rho$ decreases. Beneath the surface mixed layer, $\kappa_\rho$ is highest between 400 and 500 m ($< 10^{-4.5}$ m$^2$ s$^{-1}$; Fig. 7a), with low values predominating in the core of the high-salinity SUW ($< 10^{-6}$ m$^2$ s$^{-1}$; Fig. 7a).

Vertical turbulent heat and salt fluxes, $Q_h$ (W m$^{-2}$) and $Q_S$ (kg m$^{-2}$ s$^{-1}$) respectively, can be calculated from $\kappa_\rho$:

$$Q_h = -\rho C_p \kappa_\rho \Theta_z \tag{8}$$

$$Q_S = \frac{1}{1000}(-\rho \kappa_\rho S_z) \tag{9}$$

where $\rho$ is density, $C_p$ is the specific heat capacity of seawater, $\Theta_z$ is the vertical gradient of conservative temperature, and $S_z$ is the vertical gradient of absolute salinity.

Beneath the surface mixed layer, both $Q_h$ and $Q_S$ are predominantly negative (i.e. downward) because temperature and salinity decrease with depth (Fig. 4a and b). The most prominent feature of the distributions of both is the broad region of negative (i.e. downward) turbulent heat and salt transport between approximately 200 and 500 m (Fig. 7b and c). This corresponds to the elevated values of $\kappa_\rho$ ($> 10^{-5}$ W kg$^{-1}$) found within same depth range (Fig. 7a). Within the surface mixed layer, notwithstanding the limited coverage of the observations, $Q_h$ is positive in the top 50 m and negative between 50 and 100 m; $Q_S$ is positive throughout the surface mixed layer (Fig. 7b and c). This is due to weak thermal and haline inversions near the surface; the depths of the temperature and salinity maxima are indicated by the black lines in Fig. 7b and c respectively.

We now focus on the 200 to 500 m depth range, the region of highest $\kappa_\rho$ and, consequently, the region in which $Q_h$ and $Q_S$ are most pronounced. Over the period of the observations, the arithmetic mean $Q_h$ between 200 and 500 m was $-1.40$ W m$^{-2}$. This, and all subsequent flux estimates, are summarised in Table 1. The arithmetic mean $Q_S$ between 200 and 500 m was $-5.84 \times 10^{-8}$ kg m$^{-2}$ s$^{-1}$. This is a relatively low-turbulence region; the attendant turbulent fluxes are correspondingly relatively small and likely have little influence on the region's hydrography. For instance, integrated over a year, $Q_h$ results in an annual turbulent heat flux of $-4.43 \times 10^7$ J m$^{-2}$, which would reduce the temperature of the overlying SUW layer (assumed to be 100 m thick) by just $0.11°$C. Similarly integrated over a year, $Q_S$ results in an annual turbulent salt flux of $-1.84$ kg m$^{-2}$, which would reduce the salinity of the overlying SUW layer by just $0.02$ g kg$^{-1}$.

### 3.3 Salt fingers and associated fluxes

Diffusivity as estimated using the Osborn relation, $\kappa_\rho$ (Eqn. 7) and the associated fluxes presented above are derived from $\varepsilon$, and so are applicable to transports of heat and salt that are driven by turbulent, mechanical mixing (turbulent regime). However, these $\varepsilon$-based estimates do not account for the fluxes driven by the salt fingers (i.e. double-diffusive mixing; salt-finger regime) that are characteristic of the thermohaline staircases prominent in the western tropical Atlantic (Schmitt et al., 1987; Rollo et al., 2022); Seaglider 620 was deployed at the edge of the region identified by Schmitt et al. (1987) as being the location

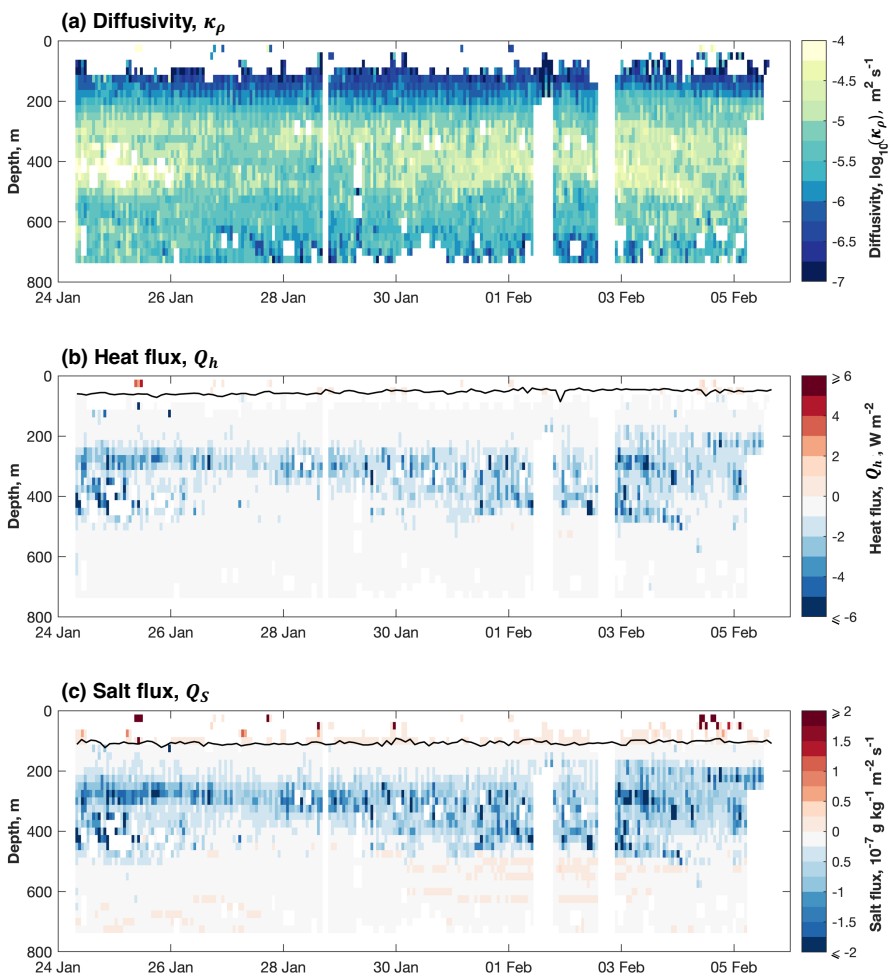

**Figure 7. (a)** Diffusivity, $log_{10}(\kappa_\rho)$ ($m^2 \, s^{-1}$). **(b)** Vertical heat flux, $Q_h$ ($W \, m^{-2}$). **(c)** Vertical salt flux, $Q_S$ ($10^{-7} \, g \, kg^{-1} \, m^{-2} \, s^{-1}$). Negative fluxes are downward. All are calculated from $\varepsilon$ calculated using the Batchelor spectrum method. The black lines in panels (b) and (c) indicate the depth of, respectively, the temperature and salinity maxima, as derived from 1 m binned Seaglider observations.

of strong staircase structures. Such structures are prominent in the temperature observations (Fig. 8). In the turbulent regime, thermal diffusivity, $\kappa_\Theta$ and haline diffusivity, $\kappa_S$, are the same. But in the salt finger regime, $\kappa_S$ may be approximately twice 255 $\kappa_\Theta$, hence they shall hereafter be considered separately. Moreover, in low-turbulence regions, such as that of the present study, salt fingers give rise to fluxes of heat and salt that can be larger than those driven by mechanical turbulence (Schmitt, 1988).

Those regions of the water column that are susceptible to salt fingers may be identified using the Turner angle, $Tu$: salt fingers can occur where $45° < Tu < 90°$. The following equations are applied only where this condition is met. $Tu$ is calculated from temperature and salinity binned into 5 m vertical bins; all subsequent diffusivities are calculated from variables binned into 260 5 m vertical bins in order to match $Tu$.

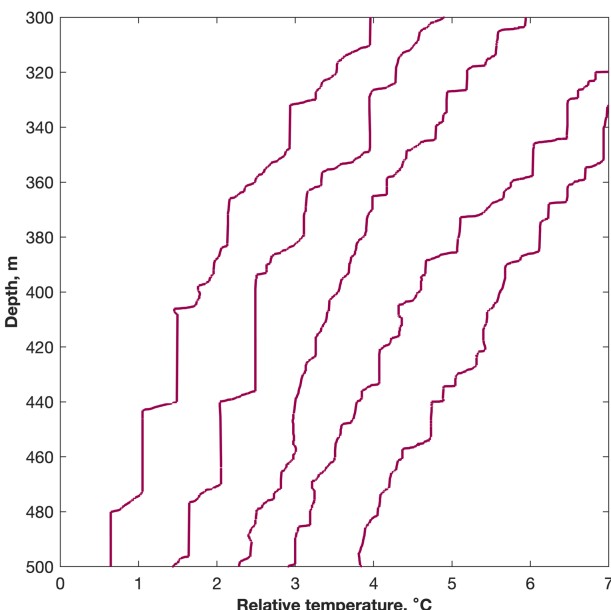

**Figure 8.** A selection of temperature profiles (presented as relative conservative temperature, °C) recorded by the FP07 fast thermistor. Thermohaline staircases are visible at a variety of scales.

We compare two different estimates of $\kappa_\Theta$ and $\kappa_S$ in the salt finger regime. Firstly, and given that theoretical flux laws can overestimate $\kappa_S$ in the real ocean (e.g. Taylor and Veronis, 1996; Kelley et al., 2003; Radko, 2005), we follow van der Boog et al. (2021) in using the empirical relations of Radko and Smith (2012) to calculate $\kappa_\Theta$ and $\kappa_S$ from the density ratio, $R_\rho$ (Eqn. 6), and the molecular diffusivity of heat, $K$, estimated from temperature and salinity at each grid point:

$$\kappa_S = \left( \frac{135}{(R_\rho - 1)^{1/2}} - 62.75 \right) K R_\rho \tag{10}$$

From $\kappa_S$, again following van der Boog et al. (2021) and Radko and Smith (2012), we then calculate $\kappa_\Theta$:

$$\kappa_\Theta = \kappa_S \frac{\gamma}{R_\rho} \tag{11}$$

where $\gamma = 2.709e^{-2.513R_\rho} + 0.5128$ is the density flux ratio in the presence of salt fingers. We refer to these salt finger diffusivities as $\kappa_\Theta$ and $\kappa_S$ from $R_\rho$ (Figure 9a and b).

Secondly, we use the Osborn and Cox (1972) relation to estimate $\kappa_\Theta$ and $\kappa_S$ using $\chi$ (Eqn. 2; see also e.g. St Laurent and Schmitt, 1999; Schmitt et al., 2005; Ijichi and Hibiya, 2018), which we calculated as an intermediary step in the calculation of $\varepsilon_\mu$ (Sec. 2.2):

$$\kappa_\Theta = \frac{\chi}{2\Theta_z^2} \tag{12}$$

The relationship between $\kappa_\Theta$ and $\kappa_S$ when calculated from $\chi$ remains as in Eqn. 11 (Schmitt et al., 2005; van der Boog et al.,
2021). We refer to these second salt finger diffusivities as Osborn-Cox $\kappa_\Theta$ and $\kappa_S$; we note that the Osborn-Cox relation can include a contribution of mechanical mixing on $\chi$ and hence on $\kappa_\Theta$.

There is approximately an order of magnitude difference between $\kappa_\Theta$ and $\kappa_S$ from $R_\rho$, the latter being the greater (Figs. 6c, and 9a and b). Neither distribution exhibits the same pronounced structure as $\kappa_\rho$ (Fig. 7a): whereas $\kappa_\rho$ resembles the distribution of $\varepsilon_\mu$ from which is was derived (Fig. 5a) – i.e. with elevated values between 200 and 500 m, and lower values below and,
in particular, immediately above – values of $\kappa_\Theta$ and $\kappa_S$ from $R_\rho$ are relatively constant in depth and time. Of the two, $\kappa_S$ is the larger, being generally greater than $10^{-4.5}$ m$^2$ s$^{-1}$; $\kappa_\Theta$ is generally between $10^{-6}$ and $10^{-5}$ m$^2$ s$^{-1}$, although some values higher than this are present (Figs. 6c, and 9a and b). Estimates of both $\kappa_\Theta$ and $\kappa_S$ from $R_\rho$ are higher than estimates of $\kappa_\rho$ at the same depth (Figs. 7a, and 9a and b), indicating that the vertical mixing of properties in the salt-finger regime is higher than in the turbulent regime.

We then substitute $\kappa_\Theta$ from $R_\rho$ (Fig. 9a) into Eqn. 8 in place of $\kappa_\rho$ (Fig. 7a). Averaged (arithmetic mean) between 200 and 500 m, $\kappa_\Theta$ from $R_\rho$ gives rise to a heat flux of $-1.71$ W m$^{-2}$ in the salt-finger regime (Table 1), an annual temperature reduction of $0.13°$C in the SUW layer. This flux is larger than the value reported above for the turbulent regime (Sec. 3.2). Similarly, we substitute $\kappa_S$ from $R_\rho$ (Fig. 9b) into Eqn. 9 in place of $\kappa_\rho$. Averaged (arithmetic mean) between 200 and 500 m, $\kappa_S$ from $R_\rho$ gives rise to a salt flux of $-1.83 \times 10^{-7}$ kg m$^{-2}$ s$^{-1}$ in the salt-finger regime (Table 1), or an annual reduction
of $0.06$ g kg$^{-1}$ in the salinity of the SUQ layer. This flux is over three times larger than the corresponding salt flux in the turbulent regime. Note that, for the calculation of the arithmetic mean heat and salt fluxes in the salt finger regime, we set both $\kappa_T$ and $\kappa_S$ from $R_\rho$ to zero outside of the salt finger regimes, i.e. where $Tu < 45°$ and $Tu > 90°$, and include these zeros in our averages. Because we first filter out regions of the water column that are not susceptible to salt fingering, gaps in the record therefore indicate an absence of the process to be averaged (i.e. salt finger-driven fluxes) rather than an absence of data.

The distributions of $\kappa_\Theta$ and $\kappa_S$ from the Osborn-Cox relation resemble the distribution of $\varepsilon$, which itself resembles the distribution of $\chi$ that was used to calculate both diffusivities (not shown): the highest diffusivities are found between 200 and 500 m, with lower values being found above and below. In both cases, the highest diffusivities are between $10^{-6}$ and $10^{-5}$ m$^2$ s$^{-1}$; within this range, $\kappa_S$ is the greater (Figs. 6d, and 9c and d). Above and below this depth range, $\kappa_\Theta$ decreases to values below $10^{-6.5}$ m$^2$ s$^{-1}$ (Fig. 9c), whereas $\kappa_S$ decreases only to values of approximately $10^{-5.5}$ m$^2$ s$^{-1}$ (Fig. 9c).

We then substitute $\kappa_\Theta$ from the Osborn-Cox relation (Fig. 9c) into Eqn. 8 in place of $\kappa_\rho$. Averaged (arithmetic mean) between 200 and 500 m, $\kappa_\Theta$ from the Osborn-Cox relation gives rise to a heat flux of $-1.49$ W m$^{-2}$ in the salt-finger regime (Table 1), an annual temperature reduction of $0.11°$C in the SUW layer. This heat flux is very similar to that reported for the turbulent

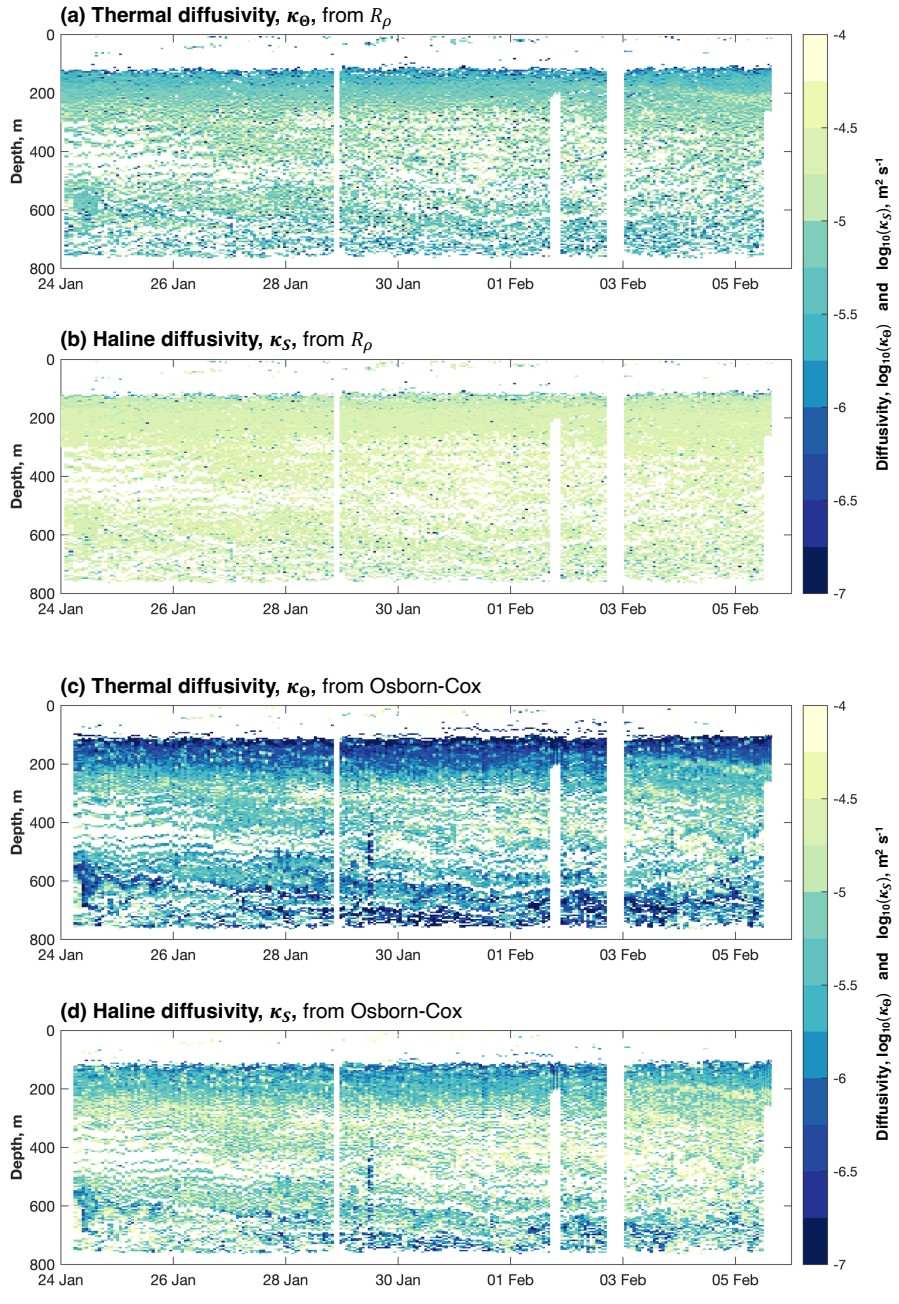

**Figure 9. (a)** Thermal diffusivity, $\kappa_\Theta$ and **(b)** haline diffusivity, $\kappa_S$ (both m$^2$ s$^{-1}$), in the presence of salt fingers as calculated from the density ratio, $R_\rho$, and the molecular diffusivity of heat, $K$ following van der Boog et al. (2021). **(c)** $\kappa_\Theta$ and **(d)** $\kappa_S$ in the presence of salt fingers as calculated from our estimates of $\chi$ using the Osborn-Cox relation (Sec. 2.2; Osborn and Cox, 1972). Diffusivities are plotted only in regions of the water column susceptible to salt fingers, i.e. where the Turner angle is between 45 and 90°.

regime, and less than that predicted by the empirical relation of Radko and Smith (2012) reported above (i.e. $\kappa_\Theta$ and $\kappa_S$ from $R_\rho$). And $\kappa_S$, averaged (arithmetic mean) between 200 and 500 m, gives rise to a salt flux of $-9.40 \times 10^{-8}$ kg m$^{-2}$ s$^{-1}$

(Table 1), an annual salinity reduction of 0.03 g kg$^{-1}$ in the SUW layer. This flux is approximately 1.6 times that reported for the turbulent regime, but half that predicted by the empirical relation of Radko and Smith (2012). As before, we set both $\kappa_T$ and $\kappa_S$ from the Osborn-Cox relation to zero outside of the salt finger regimes.

## 4 Discussion

### 4.1 Comparing $\varepsilon_\mu$ and $\varepsilon_T$

Our finding that estimates of $\varepsilon_T$ are higher than estimates of $\varepsilon_\mu$ is in agreement with the findings of Howatt et al. (2021). They report that, while the Thorpe scale method can well represent qualitative aspects of the spatio-temporal distribution of $\varepsilon$, the method can, when applied to relatively low resolution finescale observations, artificially inflate $\varepsilon_T$ by an order of magnitude relative to estimates from, for instance, the Batchelor spectrum method. This is for the simple reason that finescale observations lack the resolution to capture small-scale overturns; the resultant distribution of $L_T$, and hence of $\varepsilon$ is consequently biased

towards high values.

This is supported by our observations: the differences between $\varepsilon_\mu$ and $\varepsilon_T$ reported here are much lower, being, on average, far less than an order of magnitude (Sec. 3; Figs. 5 and 6). Further, we directly compare $L_T$ calculated using the FP07 microstructure observations (512 Hz, 100 Hz when accounting for the sensor's response time; Fig. 3) to $L_T$ calculated using the Seaglider's ordinary finescale temperature and salinity observations (0.2 Hz). The distribution of $L_T$ derived from microstruc-

320 ture observations has a pronounced peak at very low values ($> 0.25$ m; Fig. 10). In contrast, the finescale glider observations are unable to resolve values of $L_T$ at this scale (Fig. 10). The better agreement between $\varepsilon_\mu$ and $\varepsilon_T$ reported here than by Howatt et al. (2021) therefore appears to be explained by the higher vertical resolution of our observations. Consequently, and in agreement with Howatt et al. (2021), we suggest that, in a low-turbulence environment, the Thorpe scale method is unable to produce reliable quantitative estimates of $\varepsilon$ unless it is applied to microstructure observations. In a high-turbulence

environment where overturns are large, finescale observations may be better able to accurately resolve the distribution of $L_T$ and their relatively low resolution may not introduce such a systematic bias.

A potential problem with using the Thorpe scale method on Seaglider observations is the non-vertical profile that the Seaglider collects: unlike, for instance, a ship-deployed CTD or a vertical microstructure profiler, a Seaglider follows a slanted trajectory, typically covering a horizontal distance of 4 km over a 1000 m-deep dive-climb cycle. Any resultant sampling of

330 horizontal gradients, particularly in the presence of internal waves, could artificially inflate estimates of $\varepsilon_T$ due to false overturn detection (Thorpe, 2012). However, this is a concern only when the internal wave slope exceeds the slope of the Seaglider's trajectory. Following Howatt et al. (2021), we calculate that the mean ($\pm$ one standard deviation) of SG620's trajectory slope (from the horizontal) during the Eurec4a deployment was $0.70 \pm 0.09$, i.e. greater than the upper limit of the slope of internal waves (approximately 0.3; Thorpe, 1978; Howatt et al., 2021). Consequently, false overturn detection due to the Seaglider's

sloping trajectory is unlikely to lead to over-estimation of $\varepsilon_T$ in this dataset.

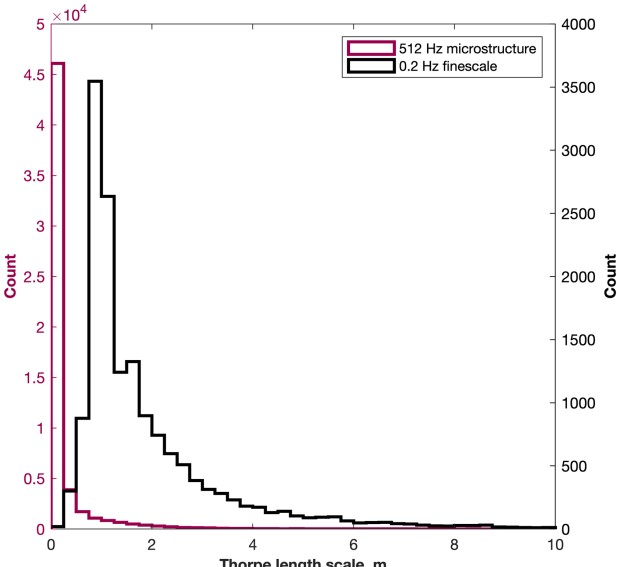

**Figure 10.** Histograms of Thorpe scale, $L_T$, as calculated from FP07 microstructure observations (green line; left-hand axis) and ordinary Seaglider finescale observations (grey line; right-hand axis).

## 4.2 Observed dissipation in spatial and temporal context

Values of $\varepsilon$ in the western tropical Atlantic are broadly consistent with other open-ocean regions away from shallow topography. For instance, Sheen et al. (2013) and Naveira Garabato et al. (2016) report background $\varepsilon$ values of approximately $10^{-10}$ to $10^{-9}$ W kg$^{-1}$ downstream of the Drake Passage, with values in excess of $10^{-8}$ W kg$^{-1}$ in the upper 1000 m and in the vicinity of rough topography. George et al. (2021) report $\varepsilon$ values between $10^{-10}$ to $10^{-8}$ W kg$^{-1}$ in the upper layers of the southwestern Bay of Bengal. And Peterson and Fer (2014) report mission-mean values of between $10^{-8}$ to $10^{-7}$ W kg$^{-1}$ from observations collected near the Faroe Islands in the northern Atlantic.

**Table 1.** Heat and salt fluxes estimates calculated using diffusivities in the turbulent and salt finger regime. Fluxes for the latter regime are estimated from $R_\rho$ using both the empirical relations of Radko and Smith (2012), and from $\chi$ using the Osborn-Cox relation (Osborn and Cox, 1972). All flux estimates have been averaged (arithmetic mean) between 200 and 500 m.

| Regime | Heat flux, $Q_h$ (W m$^{-2}$) | Salt flux, $Q_S$ (kg m$^{-2}$ s$^{-1}$) |
|---|---|---|
| Turbulent | $-1.40$ | $-5.84 \times 10^{-8}$ |
| Salt finger, $R_\rho$ | $-1.71$ | $-1.83 \times 10^{-7}$ |
| Salt finger, Osborn-Cox | $-1.49$ | $-9.40 \times 10^{-8}$ |

Limited estimates of $\varepsilon$ are available for the western tropical Atlantic. Two profiles from the region, collected by a microstructure turbulence profiler, were presented by Fernández-Castro et al. (2014). Similarly to our profiles, they find surface $\varepsilon$ values
between approximately $10^{-7}$ and $10^{-6}$ W kg$^{-1}$, and values between approximately $10^{-9}$ and $10^{-8}$ W kg$^{-1}$ between 50 and 300 m. Below 50 m, their estimates of $\kappa_\rho$ are similar to ours, although they report near-surface values several orders of magnitude larger than ours (e.g. $> 10^{-3}$ m$^2$ s$^{-1}$). Our methods do not yield reliable estimates of high turbulence in the mixed layer, however, so a full comparison is not possible. Two sets of observations taken several years apart and separated by up to hundreds of kilometres cannot be readily compared; moreover, their profiles do not extend deeper than 300 m. Interannual
variability and geographic differences could be pronounced. Nevertheless, there is no evidence in their deepest observations of an increase in $\varepsilon$ that might indicate the presence of the elevated-$\varepsilon$ layer that we observe beneath SUW (Figs. 4 and 5) and which, to our knowledge, has not previously been described.

### 4.3 Diffusivity and its influence on hydrography

The distribution of $\kappa_\Theta$ and $\kappa_S$ from the Osborn-Cox relation (i.e. from $\chi$), resembles that of $\kappa_\rho$ – itself derived from $\varepsilon_\mu$ – far
more closely than do either $\kappa_\Theta$ and $\kappa_S$ from $R_\rho$ (Figs. 7a and 9). Notably, neither $\kappa_\Theta$ nor $\kappa_S$ from $R_\rho$ seems to be particularly influenced by features of the water column that might be expected to influence vertical diffusivity, such as stratification (Fig. 4c) or temperature and salinity gradients (not shown). Given that the Osborn-Cox relation explicitly relates diffusivity to a mixing variable (i.e. $\chi$), we suspect that it is more accurate than the empirical relation of Radko and Smith (2012, i.e. $\kappa_\Theta$ and $\kappa_S$ from $R_\rho$).

As calculated from $R_\rho$ (Radko and Smith, 2012), heat flux due to salt fingers is 1.22 times greater than that due to turbulent mixing; as calculated from $\chi$ (Osborn and Cox, 1972), the heat flux due to salt fingers is 1.06 times that due to turbulent mixing. And as calculated from $R_\rho$, the salt flux due to salt fingers is 3.31 times greater that that due to turbulent mixing; as calculated from $\chi$, the salt flux due to salt fingers is 1.6 times that due to turbulent mixing. In all instances, heat and salt fluxes due to salt fingers (i.e. double-diffusive convection) are higher than turbulent fluxes. The empirical relations of Radko and Smith (2012)
predict a relatively large difference, whereas the Osborn-Cox relation predicts a difference that is relatively small – indeed, for heat, the turbulent and salt-finger fluxes are almost the same. The choice of equation used to calculate diffusivities in the salt finger regime makes a considerable difference to the result.

The dissipation rates and diffusivities presented above appear to have a limited influence on the hydrography of the study region. If the turbulent fluxes estimated above are representative of annual average conditions, heat and salt from the SUW
layer penetrate downwards into the ocean interior relatively slowly; over the 12 days of the Eurec4a Seaglider deployment, the hydrographic influence of the fluxes would be negligible. As calculated from $R_\rho$, the double diffusive heat flux due to salt fingers is 1.2 times greater than that due to turbulent mixing; as calculated from $\chi$ the double diffusive heat flux due to salt fingers is the same as that due to turbulent mixing. Salt fingers potentially have a greater influence on hydrography that turbulent mixing, but the overall effect would still be near inconsequential.

## 5 Conclusions

We demonstrate that microstructure temperature observations collected by a Seaglider-mounted FP07 fast thermistor may be used to estimate the dissipation rate of turbulent kinetic energy, $\varepsilon$. We estimate $\varepsilon$ using the Batchelor spectrum method ($\varepsilon_\mu$) and the Thorpe scale method ($\varepsilon_T$). The results from the two methods agree well, although $\varepsilon_T$ is on average higher than $\varepsilon_\mu$. This is in agreement with previous studies, although the difference reported here, which is less than an order of magnitude, is below that reported elsewhere. This improved agreement is due to $\varepsilon_T$ being calculated using the same high-resolution observations as were used to calculate $\varepsilon_\mu$: other studies (e.g. Howatt et al., 2021) have compared $\varepsilon_\mu$ to $\varepsilon_T$ calculated using lower-resolution finescale temperature observations that are unable to resolve overturns below a certain size, thus biasing estimates of $\varepsilon$ towards high values.

We identify a layer of elevated $\varepsilon$ values between 200 and 500 m that lies immediately below Subtropical Underwater, a high-salinity sub-surface water mass that is co-located with a maximum in stratification. We estimate that, over the period of the deployment, this elevated $\varepsilon$ layer is responsible for a mean heat flux of $-1.40$ W m$^{-2}$ and a mean salt flux of $-5.84 \times 10^{-8}$ kg m$^{-2}$ s$^{-1}$. Given the prevalence of double diffusion and salt fingers in the region, we estimate thermal and haline diffusivities in the salt-finger regime, and the resultant heat and salt fluxes using both the empirical relation of Radko and Smith (2012), which depends on the density ratio, $R_\rho$, and the relation of Osborn and Cox (1972), which depends on the rate of destruction of temperature variance, $\chi$. Fluxes estimated using the relation of Radko and Smith (2012) are higher than those estimated using the relation of Osborn and Cox (1972). Nevertheless, turbulent and salt finger fluxes are small, and their influence on hydrography is likely limited.

*Code availability.* The Matlab toolbox of Scheifele et al. (2018) to calculate turbulent kinetic energy dissipation rate following the Batchelor spectrum method is available at: github.com/bscheife/turbulence_temperature.

*Data availability.* Standard hydrographic observations from SG620 (Rollo, 2021) are available from the British Oceanographic Data Centre at: *doi:10.5285/c596cdd7-c709-461a-e053-6c86abc0c127*. Processed turbulent kinetic energy dissipation rate estimates (Rollo et al., 2023), from both Batchelor and Thorpe scale methods, are available from the British Oceanographic Data Centre at: *doi:10.5285/f173b9c1-bb50-0b75-e053-6c86abc02a4a*.

*Author contributions.* PMFS and GMD calculated turbulent kinetic energy dissipation rate following the Batchelor spectrum method. PJL calculated turbulent kinetic energy dissipation rate following the Thorpe scale method. All authors contributed to the analysis. PMFS wrote the paper with assistance from GMD and PJL, and with comments and feedback from KJH and RAH.

*Competing interests.* The authors declare that they have no competing interests.

*Acknowledgements.* PMFS, GMD, KJH and RAH were supported by the COMPASS project (grant number 741120), which was funded by the European Research Council under the Horizon 2020 programme. PJL was funded by NERC-UKRI NEXUSS PhD studentship NE/N012070/1. We thank Beth Siddle, Callum Rollo, and the crew and scientists of the RV Meteor (cruise M161) for assistance in the deployment and recovery of the glider. We thank Gareth Lee and Marcos Cobas-Garcia for the preparation of the glider, and the UEA glider group for piloting. We are indebted to two anonymous reviewers, whose feedback has greatly improved this paper.

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
