# Peer review of "Turbulent kinetic energy dissipation rate and associated fluxes in the western tropical Atlantic estimated from ocean glider observations"

_EGUsphere, 2022_

## Referee Comment (RC2)

Review of the manuscript entitled "Turbulent kinetic energy dissipation rate and associated fluxes in the western tropical Atlantic estimated from ocean glider observations" by Peter M.F. Sheehan et al., submitted to EGUsphere

Overall:

This paper described turbulent energy dissipation rate $\varepsilon$ obtained by glider observations from two methods: one is estimated by estimating Batchelor wavenumber using fast-response thermistor and the other is from Thorpe-scale density overturn. The former $\varepsilon$ is less than the latter one. The turbulent heat and salt fluxes are reported not to be influential, while the salt-finger double diffusive fluxes could be influential. These results are potentially worth to be published, but need to be further analyzed, because both the discussions why the difference between the two methods and for the salt-finger fluxes are not enough.

Specific points:

1. Methods of estimating the $\varepsilon$ is not enough. Time response of the fast-response thermistor probe is not enough and needs to be corrected even for the glider observations. The underestimation from the spectrum fit could be caused by insufficient spectrum correction. Also the original method for estimating Batchelor wavenumber was proposed by Oakey (1982) and Ruddick et al. (2000) which should be cited.

2. The regression coefficient between the Thorpe and Ozmidov -scales are also not decisive, and the Thorpe-scale method cannot estimate $\varepsilon$ for depths without density inversions. The standard method for estimating $\varepsilon$ is shear-probe measurements. Before comparing $\varepsilon$ between the Batchelor and Thorpe methods, comparisons between shear and Batchelor and between shear and shear and Thorpe methods should be discussed or appropriate previous studies should be cited.

3. The estimation for the salt-finger double diffusive fluxes is relied on the parameterization for very simple situation. Discussion using the microstructure measurements should be done to evaluate the validity of the parameterization. The estimate of the density ratio $R_\rho$ is quite important for the parameterization. The procedure (bin length for gradient estimates) should be further described.

4. Discussion is necessary on whether the estimated double diffusive mixing explains water property changes during the present 13 days observations.

Minor points:

■ Figure 4: figure for $\log10(\varepsilon_T)$ versus $\log10(\varepsilon_\mu/\varepsilon_T)$ is desirable. From Fig.4c, the

differences are both plus and minus.

- 195th : Fig. 5a → Fig. 5 ?

- 208th : ($<10^{-4.5}$     ) → ($>10^{-4.5}$     ) ?

- 217th : $\varepsilon$ → $K_\rho$ ?

---

## Author Comment (AC1)

**Response to reviewer 1**

Throughout this document, the two reviews are reproduced in black. Our responses are presented in blue and, where appropriate, quotations from the revised paper are included in indented *italic blue* text.

**REVIEWER 01**

In this manuscript, Sheehan et al. describe results from a Seaglider mission in January-February 2020 in the western tropical Atlantic. The paper focuses on reporting dissipation rates of turbulent kinetic energy using two methods: a direct microstructure method, with which temperature fluctuations are measured at centimetre-scales, and an indirect method based on the meter-scale vertical overturns detected in the temperature profiles (the so-called Thorpe method), also measured with a microstructure probe. The most interesting pattern in the observations is a local enhancement of turbulent dissipation below a subsurface layer of high-salinity associated with the subtropical underwater. The authors claim that this is the first study reporting microstructure temperature measurements from this particular type of glider. Another highlight of the paper is that, contrary to some previous assessments, the direct and indirect methods for estimating dissipation rates agree well in their dataset. Finally, the authors also report and discuss heat and salt fluxes driven by the measured turbulent motions and compare them with those potentially caused by double-diffusive instabilities, for which the conditions are favourable in the region.

Although the paper has some strong points (it describes novel dataset with some interesting results, it is well written), it has also – in my view – some major weaknesses that need to be addressed before publication. My general feeling is that the authors missed several good opportunities to make the paper more relevant and useful for their colleagues. I list my comments below with some suggestions of further analysis/discussions that could potentially improve the impact of the article:

We thank the reviewer for recognising the value of the dataset and the work, and for the thorough suggestions they have made regarding its improvement. We agree that they will increase the impact of the article.

(1) Microstructure Method. Overall, I missed a more extensive and direct acknowledgement of the limitations of the different methods used. The authors use a nowadays standard method to estimate energy dissipation rates from temperature microstructure measurements (the Batchelor fitting method). However, from my experience and the available literature, the method has important limitations and the results may be highly dependent on some user choices. In my opinion, those aspects are relevant to discuss. One key limitation is the fact that the dissipation rates detectable with the method are upper bounded due to the limited time response of the FP07 sensor. For typical glider along-path speeds of 40 cm/s, maximum dissipation rates measurable are about 10 W/Kg. The authors report values larger than that in the upper mixing layer, but I haven't seen any mentioning as to how the sensor limitation may affect their results, in particular those values, and the comparison with the Thorpe-scale method. I would like to see some of the temperature-gradient spectra in the upper mixing layer and the fit parameters, to confirm that those values correspond to real dissipation. My experience is that for high

epsilon values the roll-off in the temperature gradient is probably imposed by sensor limitations, such that the Batchelor fitting is not really meaningful (see also Paterson and Fer 2014).

We thank the reviewer for bringing this limitation to our attention. Following Peterson and Fer (2014), we will remove any estimate of epsilon greater than $2 \times 10^{-7}$ W kg$^{-1}$. We will add this as a quality control criterion in Section 2.2:

> *"The estimate of $\varepsilon_\mu$ is greater than $2 \times 10^{-7}$ W kg$^{-1}$; even having corrected estimates of $\Lambda_{32}$ at high frequencies, the effective resolution of the fast thermistor (100~Hz) is such that values of $\varepsilon_\mu$ greater than this cannot be reliably estimated."*

Furthermore, we will outline the expected influence of sensor limitation on our results in Section 3.1:

> *"But these highest values [$\varepsilon_\mu > 10^{-8}$] are infrequently observed [in the surface mixed layer]: a large proportion of $\varepsilon_\mu$ estimates in the mixed layer are greater than $2 \times 10^{-7}$ W kg$^{-1}$, beyond the range for which the FP07 fast thermistor observations and the Batchelor spectrum method yield meaningful results (Sec. 2.2; Peterson and Fer, 2014). Consequently, the remaining values are all below this threshold; indeed, considerable variability in $\varepsilon_\mu$ is observed in the mixed layer, and many remaining estimates are low ($< 10^{-10}$). The mean $\varepsilon_\mu$ in the upper water column is therefore likely to be biased towards these low values."*

The time-depth plot of estimates of $\varepsilon_\mu$ with the addition of the new upper-limit cut off ($\varepsilon_\mu > 2 \times 10^{-7}$ W kg$^{-1}$) are presented in (Figure 1a) at the end of this document. Examples of relatively high $\varepsilon_\mu$ ($10^{-8} < \varepsilon_\mu < 2 \times 10^{-7}$ W kg$^{-1}$) spectra from the mixed layer are presented in (Figure 2) at the end of this document. We note that estimates of $\varepsilon_\mu$ within this range are few, given that many already fail the quality control procedure of Scheifele et al (2018) and Scott et al (2021).

Further, the temperature spectra are typically partially corrected for the sensor time-response. The best parameters for the adjustment are not well stablished and can differ from sensor to sensor. It would be good to report explicitly what exact correction is used here.

We apologise for omitting this information in our summary of the method of Scheifele et al. (2018). Once the temperature power spectrum, $\Lambda_{32}$, has been calculated from a 32-second segment, the transfer function of Sommer et al. (2013) is used to correct for the slow FP07 response time at high frequencies. This step is included in the Matlab toolbox provided by Ben Scheifele and Jeffrey Carpenter. We will add this step into our summary of the method.

> *"Values of $\Lambda_{32}$ at high frequencies, where the thermal inertial of the fast thermistor is such that its temporal response is inadequate, are corrected using the transfer function of Sommer et al. (2013)."*

I also have a general vocabulary concern. The authors make a difference between what they call "spectral" methods for calculating turbulent dissipation, as opposed to non-spectral methods, like the fine-scale and Thorpe methods. In my view this is not entirely correct since, for example, the fine-scale method often involves spectral calculations. In my opinion, what distinguishes the two kind of methods is the scale at which measurements are done. In this regard, I think it would be more correct to use the term "microstructure method" as opposed to "indirect methods" (or something similar). I would recommend to change this throughout the manuscript.

We agree that, on reflection, spectral methods does not accurately and uniquely describe the methods used in this study. In the revised paper, we will use "microstructure" methods/observations to refer to the Batchelor spectrum method applied to 512 Hz FP07 temperature observations; we will use "finescale" methods/observations to refer to observations of lower resolution (e.g. ship and glider CTD) and to the methods commonly used to process them (e.g. Thorpe scale, finescale parameterisation).

(2) Thorpe-scale method. Regarding the Thorpe method, its applicability to non-vertically profiling instruments like gliders has been questioned in the past (e.g. Thorpe et al. 2012), since the method is based on measuring the vertical size of turbulent overturns, which is not entirely achievable with a slant-wise measuring trajectory. Although your results are encouraging, I still think it would be better to acknowledge and discuss the potential impact of this limitation more explicitly.

We thank the reviewer for highlighting this potential limitation of using the Thorpe scale method on glider observations. Thorpe (2012) suggested that internal waves, when present, can lead to false overturn detection and therefore to artificially high estimates of $\varepsilon_T$; Howatt et al. (2021) further suggest that the detection of false overturns is of concern when the slope of internal waves is greater than the glide slope (i.e. angle from the horizontal) of the glider. Following Howatt et al. (2021), we calculate that the mean (± one standard deviation) glide slope for the present mission is 35 ± 5 degrees – i.e. greater than the maximum slope of internal waves (16 degrees). We will add an acknowledgement of this to the revised paper:

> *"A potential problem with using the Thorpe scale method on Seaglider observations is the non-vertical profile that the Seaglider collects: unlike, for instance, a ship-deployed CTD or a vertical microstructure profiler, a Seaglider follows a slanted trajectory, typically covering a distance of approximately 4 km over a 1000 m dive-climb cycle. Any resultant sampling of horizontal gradients, particularly in the presence of internal waves, could artificially inflate estimates of $\varepsilon_T$ due to false overturn detection (Thorpe 2012). However, this is a concern only when the internal wave slope exceeds the slope of the Seaglider's trajectory. Following Howatt et al. (2021), we calculate that the mean (± one standard deviation) of SG620's trajectory slope (from the horizontal) during the Eurec4a deployment was 35 ± 5°, i.e. greater than the upper limit of the slope of internal waves (16°; Thorpe, 1978). Consequently, false overturn detection due to the Seaglider's sloping trajectory is unlikely to lead to over-estimation of $\varepsilon_T$ in this dataset."*

Also, the authors contrast their results (in which the microstructure and Thorpe methods agree well), with a previous study (Howatt et al. 2021) in which the Thorpe method tended to overestimate dissipation. They speculate that this difference may be due to the different probe resolution: in the present study a microstructure probe is used, which allows a much better resolution of the smaller overturns, whereas Howatt et al. (2021) use a standard CTD probe. While this is just a mere speculation, the authors have the available data to evaluate this hypothesis: they could apply the Thorpe method to the glider regular CTD data and compare the results with those of the FP07-Thorpe method. I think not doing that is a missed opportunity, since it would be very useful for future studies to know whether the disagreement comes from the different spatial resolution or not.

We thank the reviewer for this suggestion. We feel that the original wording in our discussions did not fully convey the weight of the findings of Howatt et al. (2021), who clearly conclude that the low resolution of a glider CTD sets a minimum resolvable turbulent patch size. In oceanographic conditions where overturns are likely to occur over much smaller scales than the minimum resolvable turbulent patch size, ensuing estimates of $\varepsilon_T$ will necessarily be biased high. We agree with this argument. Our original discussion should have made this clearer. Values of LT derived from the FP07 fast thermistor observations (512 Hz) are clearly far smaller than those derived from ordinary Seaglider observations

(0.2 Hz; Figure 3 of this document), indicating that the latter are unable to resolve the majority of overturns in this region. Estimates of $\varepsilon_T$ from these ordinary Seaglider observations will therefore be extremely unreliable. Given that Howatt et al. (2021) have already reported this conclusion, we do not want to repeat their work and include figures illustrating this difference, but we will comment on the inability of the ordinary Seaglider observations to accurately resolve $L_T$.

I have another concern about the method, in particular about what you describe in line 145 regarding the aggregation of overturns until they reach a scale of 2 m. Sorry, I am not familiar with this approach, but it seems a bit weird to me, why do you need to impose a minimal overturning scale if those appear smaller in the data? This could bias high the lower dissipation rates, couldn't it?

We thank the reviewer for bringing this to our attention. The aggregation of overturns is traditionally done during Thorpe scale calculations with relatively low-resolution observations in order to ensure that there are sufficient data points within each turbulent patch (e.g. Howatt et al., 2021). We originally included this step, despite having very high-resolution observations, for consistency with previous work. However, we are re-examining whether this step is indeed necessary when using 512 Hz observations, and whether it makes an appreciable difference to estimates of $\varepsilon_T$. We will include this in the revised paper.

(3) Salt fingers. I have two main comments regarding the double diffusion diffusivity calculations and fluxes. First, I think there is a clear missed opportunity here of using the microstructure data to assess the role of double diffusion more directly. I missed, for example, a figure showing some of the staircases, but more importantly, you could use the thermal variance dissipation rates to directly assess the rate of heat mixing in the salt-finger favourable zone using the Osborn-Cox method. Several authors (e.g. St Laurent et al. 1999; Schmitt et al. 2005) have reported levels of thermal variance dissipation in thermohaline staircases, and heat diffusivities above those predicted by the Osborn model (your eq. 7). In most microstructure studies based on shear probes this information is missing, but you have it and you could compare the results with the flux law estimates.

As discussed immediately below, we will change the method used to identify those regions susceptible to salt fingering; nevertheless, we will happily include a figure illustrating salt fingers if it adds to the arguments of the revised paper. We agree that an estimate of heat diffusivity based on our estimates of $\chi$ would make for an interesting comparison with the existing results. We will include such an estimate in the revised paper, likely as an additional panel in Figure 8.

My second comment concerns the latter method. I see that in your Figure 7, salt fingers diffusivities are zero except in the step layers, however it is not clear how you deal with those zeros in the calculation of mean diffusivities and fluxes. If you just neglect them, you may be overestimating the average diffusivities by quite a lot! But this is not explicit. Can you clarify that? That said, I am not entirely sure about the way in which the flux-law formulae should be applied, but my intuitive guess would be that the equations should be applied for the entire region of the vertical profile where the density ratio is favourable for salt fingers and the steps are observed, that is, applied to the background temperature-salinity gradients instead of the gradients in the steps, as you did. Not sure if I am wrong on that, but I invite you to dive a deeper into it and find out what is the most correct way of applying the formulae and report the details (how and where the diffusivities, gradients and fluxes were calculated; how did you deal with zeros.

Having more closely studied the original paper by Radko and Smith (2012), we agree with the reviewer that Equations 10 and 11 should be applied to the entire region of the vertical profile where the density ratio is favourable for salt fingers, rather than just in the gradient layers. Consequently, we will use the

Turner angle to identify those parts of the water column that are susceptible to salt fingering, and will apply the equations to these regions. Further, we agree that it would be more accurate to calculate a time-depth average while including regions in which the fluxes of heat and salt due to salt fingering are zero – i.e. to include these zeros in our averages. We thank the reviewer for making these suggestions.

(4) Discussion of the observed patterns. The main feature of your dissipation dataset is an enhancement in a thick layer below the salinity maximum. In my view, there is also a missed opportunity here of making this result more meaningful and relevant by discussing the possible drivers of this enhancement. Is that maybe due to internal wave interactions with the rapidly changing stratification? Do you have current measurements from the ship that indicate a local enhancement of shear or other data that could give some clue of what is going on? Not sure if further analysis on this is needed for the manuscript (which has a strong methodological component), but I think it could help to make the paper more relevant and interesting.

As the reviewer suggests, we will compare our results with ADCP observations from the ship that passed closest to the glider during the deployment, and will comment on this in the revised paper.

SPECIFIC COMMENTS:

Line 8. Superscript missing in W kg$^{-1}$

This will be corrected.

Line 21. The Batchelor method applies only to scalars (e.g. temperature), not shear. The use in this sentence is not correct

This sentence will be altered from:

*"Estimating the dissipation rate of turbulent kinetic energy, ε, by applying the Batchelor spectrum method (Batchelor, 1959) to high-resolution observations of shear and temperature (e.g. Lueck et al., 2002; Peterson and Fer, 2014; Scheifele et al., 2018) has, historically, required considerable ship time, plus specialist instruments and expertise."*

to:

*"Estimating the dissipation rate of turbulent kinetic energy, ε, from high-resolution observations of shear and temperature (e.g. Lueck et al., 2002; Peterson and Fer, 2014; Scheifele et al., 2018) has, historically, required considerable ship time, plus specialist instruments and expertise."*

Line 25. These authors report 266 profiles in 50 different stations

We will remove this statement.

Line 61, the use of "," and "()" in this sentence, and other locations in the manuscript, seems weird to me

We will make changes to improve punctuation in various locations in the manuscript.

Line 73. Why didn't you use the shear probe data in this paper? It feels also like a potential missed opportunity of a more thorough evaluation of the different methods.

There was a fault with the shear probe and the data are unfortunately unusable. This will be mentioned in the revised paper:

*"There was a fault with the shear probe on this deployment and the observations could not be used."*

Line 95. This sentence "the rate of destruction of temperature-gradient variance" is not correct: note that chi units are K^2 per second, not (K/m)^2 per second. Temperature of thermal variance is destroyed, not the temperature-gradient variance. Remove "-gradient"

This will be corrected as suggested.

Line 114—115. I am not familiar with this paper but the normalisation of the temperature gradient spectra with epsilon seems a bit unphysical to me, since the spectrum scales with chi not with epsilon. There is a proper way of normalising the temperature variance spectrum (see e.g. Dillon and Caldwell (1980)).

The full procedure developed by Scott et al. (2021) is described in the supplementary material to the paper we cited (available at *agupubs.onlinelibrary.wiley.com/doi/full/10.1029/2020JC016861*). We normalise estimates of $\varepsilon$ only to perform this quality control step: it is the original, non-normalised values of $\varepsilon$ that are subsequently considered in the paper.

Line 123. You could perhaps define the meaning of the term (epsilon/N)^1/2. Also note the wrong spelling in "Tayor"

This sentence will be altered from:

*"If the quantity $U/(\varepsilon_\mu /N)^{1/2}$ is less than five, where U is the glider's speed-in-direction-or-travel, indicating that Tayor's frozen turbulence hypothesis is invalid."*

to:

*"If the quantity $U/(\varepsilon_\mu /N)^{1/2}$ is less than five, where U is the glider's speed-in-direction-of-travel and $(\varepsilon_\mu /N)^{1/2}$ is an estimate of the turbulent flow velocities (Fer et al., 2014), indicating that Taylor's frozen turbulence hypothesis is invalid."*

Line 161 onwards. How was the critical value chosen? It seems a bit arbitrary. Why don't you use the alpha and beta coefficients to directly estimate the contributions of T and S gradients/variability to density stratification/variability? A more detailed description of the results of this quality check would be desirable. How many datapoints have been discarded? Where?

We thank the reviewer for bringing this to our attention. We agree that using alpha and beta coefficients (i.e. the stability ratio) would be more appropriate as a quality control measure. In the revised paper, we will implement a quality control check based on the stability ratio, and we will provide a detailed description of the impact of this on our dataset.

Figure 3 or 4. It could be nice to see a distribution of LT here or elsewhere.

We agree that this would make a nice addition to the paper. A time-depth section of $L_T$ is presented at the end of this document (Figure 4) and will be included in the revised paper.

Figure 4. Why are there no epsilon_T values in the upper mixing layer? What is the criterion to discard those values? (I guess it is related to the very weak stratification but you should mention it)

Values of $\varepsilon_T$ were removed above 75 m, as there is an apparent large overturn in temperature at approximately 60 m depth. From the glider CTD data, it is clear that this is density-compensated and is therefore not a turbulent overturn. This will be added to the text:

> *"We remove all values of $\varepsilon_T$ shallower than 75 m because there is an apparent large temperature overturn above this depth that is, in fact, density-compensated; hence, this is not a real overturn."*

Section 3.1. Uncertainties reported in this section have inconsistent values and units, also with respect to Figure 4. E.g. values in line 180, are much higher than those in 184 (and have no units, while those in 184 do have units), and they also seem different from the error margins in the figure. Please, clarify how the standard deviations are calculated and improve the consistency.

We thank the reviewer for pointing out these inconsistencies, in particular the incorrect use of units when discussing the dimensionless geometric standard deviation factor in line 184. We had not included a definition of the geometric standard deviation factor in the manuscript as this can be found in standard mathematical textbooks. However, given that many readers will not be familiar with geometric means and geometric standard deviation factors, we will expand the descriptions given here to provide some clarification.

Lines 131 to 134 will be altered from:

> *"Quality-controlled estimates of $\varepsilon_\mu$ were binned, profile by profile, into 25 m bins; we use the geometric mean (and standard deviation) in preference to the arithmetic mean, the better to represent the average (and spread) of observations that span many orders of magnitude."*

to:

> *"Quality-controlled estimates of $\varepsilon_\mu$ were binned, profile by profile, into 25 m bins; we use the geometric mean in preference to the arithmetic mean, the better to represent the average of observations that span many orders of magnitude and which are not normally distributed."*

Lines 179 to 180 will be altered from:

> *"The greater noise in $\varepsilon_T$ is reflected in its having a higher standard deviation factor (5.57) than $\varepsilon_\mu$ (3.94). (Note that geometric standard deviation factor is multiplicative, not additive.)"*

to:

> *"The greater noise in $\varepsilon_T$ is reflected in its having a higher geometric standard deviation factor (GSDF) of 5.57 than $\varepsilon_\mu$, the GSDF of which is 3.94. (Note that GSDF is multiplicative, not additive, and is therefore dimensionless. The one-standard-deviation-range is from the geometric mean/GSDF to geometric mean × GSDF.)"*

Throughout the paper, we have clarified when we are using the arithmetic or geometric mean and standard deviation.

Line 195. There is no panel a in Figure 5.

This will be altered to simply refer to Figure 5.

Lines 195—196. Not sure I see this modest peak you talk about, or maybe it is too modest to be worth mentioning?

This sentence will be removed: we agree that the modest peak is very hard to see.

Line 199 – 200. Is there something wrong with this sentence? I do not get it.

The wrong number was used to illustrate this point. This will be corrected:

*"This is in contrast to higher values of $\varepsilon_\mu$ and $\varepsilon_T$ (> $10^{-10}$ W kg$^{-1}$) within the 700 to 800~m depth range at the beginning and end of the deployment."*

Line 205. Why? (see comment above)

As mentioned above, values of $\varepsilon_T$ were removed above 75 m, as there is an apparent large overturn in temperature at approximately 60 m depth.

Line 206—207. I think you need a bit more clarity in this sentence. What makes the epsilon and kappa distributions similar?

This will be changed from:

*"The distribution of $\kappa_\rho$ resembles that of $\varepsilon_\mu$, notwithstanding a decrease in $\kappa_\rho$ at mid-depth between the 26 and 28 January where $\varepsilon_\mu$ remains relatively high (Figs. 4a and 6a)."*

to:

*"Below approximately 250 m, the distribution of $\kappa_\rho$ resembles that of $\varepsilon_\mu$ due to the relatively low variability in $N^2$ at these depths (Fig. 3c). Above 200 m $N^2$ increases substantially in the pycnocline and thus $\kappa_\rho$ decreases."*

Line 216. Epsilon units missing

Units will be added.

Line 217. Is there an upward diffusive heat flux in the upper layer? Does that mean that thermal stratification is inverse?

There is indeed a small thermal inversion near the surface – i.e. the temperature maximum (> 28°C) is found at a depth of approximately 60 m. Above this, temperature decreases by approximately 0.5°C. This is not visible in Figure 3a given the contour interval. We will explain this in the revised paper in relation to the upward heat flux visible in Figure 6b.

Line 235. "[...] and given that theoretical flux laws can overestimate κS in the real ocean [...]". This sentence needs a reference.

References to Taylor and Veronis (1996), Kelley et al. (2003) and Radko (2005) will be added in the revised paper.

Line 245. By eye this difference looks greater. More details on how the averaging is performed are needed.

We will be changing the method by which we selected those parts of the water column in which we calculate the fluxes of heat and salt due to salt fingering (see above), so these figures and average numbers are going to change. We will provide more details on how the averages are performed – in particular including zeros in between salt fingering regions.

Line 258. "sampling and the necessary setting of a minimum Thorpe scale", why do you need a minimum Thorpe scale? (see comment above)

We apologise, the reference to setting a minimum Thorpe length scale was an error: the original text should have referred to the setting of a minimum length scale for a turbulent patch. The discussion of the influence of resolution on the reliability of $\varepsilon_T$ estimates will be thoroughly revised, but the original error will be corrected.

Lines 262-263. "however, the lack of similarly high-resolution salinity observations means that the resolution of N used to calculate εT in Eqn. 5 will limit any improvement that a small minimum LT might otherwise have on estimate of εT". The N that you introduce in the εT formula is, in my view, a background N representative of scales larger than the turbulent overturns, so I don't necessarily see the lower resolution of N as a problem here.

We agree with the reviewer that the N that we introduce in the $\varepsilon_T$ formula is meant to be a background N representative of scales larger than the turbulent overturns. Therefore, the lack of high-resolution salinity observations is not a problem. We will remove this sentence from the revised paper.

Lines 264-265. "Howatt et al. (2021), who reduce the difference between their two estimates of ε," I don't get the meaning of this sentence

This comment will be removed.

Line 285. See main comment (4)

Please see our response to main comment (4)

Lines 290-292. "The downward flux of heat due to double diffusion is almost four times that due to turbulent mixing; the downward flux of salt due to double diffusion is 7.5 times that due to turbulent mixing.". The differences in diffusivities (as reported in line 245), appear much weaker. More details are needed on this calculation.

The depth-time distributions of thermal ($K_\Theta$) and haline ($K_S$) diffusivity do not match the depth-time distribution of turbulent diffusivity ($K_\rho$). One can imagine, therefore, a point with a large temperature gradient, $\Theta_z$, a large $K_\rho$ but a small $K_\Theta$; and another point with a small $\Theta_z$, a small $K_\rho$ but a large $K_\Theta$. The

ratio of average $K_\rho.\Theta_z$ to average $K_\Theta.\Theta_z$ will not be the same as the ratio of $K_\rho$ to $K_\Theta$. We will make this point clearer in the revised paper.

Lines 298-300. "We posit that this improved agreement is due to $\varepsilon T$ being calculated using the same high-resolution observations as were used to calculate $\varepsilon\mu$: other studies have compared $\varepsilon\mu$ to $\varepsilon T$ calculated using lower-resolution temperature observations." That seems plausible but you have the data to test this and I think you should (see main comment (2))

Please see our response to main comment (2).

[Figure]

**Figure 1.** Turbulent kinetic energy dissipation rate, ε (W kg⁻¹), as estimated using **(a)** the Batchelor spectrum method ($\varepsilon_\mu$) and **(b)** the Thorpe scale method ($\varepsilon_T$). The respective means (thick lines) and standard deviations (thin lines) are shown in the panels on the right. Note that geometric standard deviations is multiplicative. The difference between $\varepsilon_\mu$ and $\varepsilon_T$.

[Figure]

**Figure 2.** Exemplar spectra from which $\varepsilon_\mu$ was estimated. The spectrum derived from FP07 temperature observations is plotted in blue; the best-fit Batchelor spectrum is plotted in black. The polynomial fit to the observed spectra is plotted in light blue and the polynomial fit to the Batchelor spectra is plotted in red: both take the form $ax^2 + bx + c$. The parameters, as described in Section 2 of the paper, used to determine the goodness-of-fit are printed above the plot.

[Figure]

**Figure 3.** Histograms of Thorpe length scales as estimated from the FP07 fast thermistor observations (purple line) and the Seaglider's ordinary CTD observations (black line).

[Figure]

**Figure 4.** Thorpe scale, LT (m), estimated from FP07 fast thermistor observations.

---

## Author Comment (AC2)

**Response to reviewer 2**

Throughout this document, the two reviews are reproduced in black. Our responses are presented in blue and, where appropriate, quotations from the revised paper are included in indented *italic blue* text.

**REVIEWER 02**

Overall:
This paper described turbulent energy dissipation rate obtained by glider observations from two methods: one is estimated by estimating Batchelor wavenumber using fast-response thermistor and the other is from Thorpe-scale density overturn. The former is less than the latter one. The turbulent heat and salt fluxes are reported not to be influential, while the salt-finger double diffusive fluxes could be influential. These results are potentially worth to be published, but need to be further analyzed, because both the discussions why the difference between the two methods and for the salt-finger fluxes are not enough.

Specific points:
1. Methods of estimating the is not enough. Time response of the fast-response thermistor probe is not enough and needs to be corrected even for the glider observations. The underestimation from the spectrum fit could be caused by insufficient spectrum correction. Also the original method for estimating Batchelor wavenumber was proposed by Oakey (1982) and Ruddick et al. (2000) which should be cited.

We apologise for omitting information on the response time correction in our summary of the method of Scheifele et al. (2018). Once the temperature power spectrum, $\Lambda_{32}$, has been calculated from a 32-second segment, the transfer function of Sommer et al. (2013) is used to correct for the slow FP07 response time at high frequencies. This step is included in the Matlab toolbox provided by Ben Scheifele and Jeffrey Carpenter. We will add this step into our summary of the method.

> *"Values of $\Lambda_{32}$ at high frequencies, where the thermal inertial of the fast thermistor is such that its temporal response is inadequate, are corrected using the transfer function of Sommer et al. (2013)."*

Furthermore, we will be happy to include the suggested references in our revised paper.

2. The regression coefficient between the Thorpe and Ozmidov -scales are also not decisive, and the Thorpe-scale method cannot estimate for depths without density inversions. The standard method for estimating is shear-probe measurements. Before comparing between the Batchelor and Thorpe methods, comparisons between shear and Batchelor and between shear and Thorpe methods should be discussed or appropriate previous studies should be cited.

While the regression coefficient between the Thorpe and Ozmidov scales ($R_{OT}$) is indeed not constant, the work of Ijichi and Hibya (2021) demonstrates that $R_{OT}$ is not highly variable in the upper 1000 m of the water column. We feel that a full treatment of this matter is outside the scope of the present paper.

Futhermore, there was a fault with the shear probe and the data are unfortunately unusable. This will be mentioned in the revised paper:

*"There was a fault with the shear probe on this deployment and the observations could not be used."*

3. The estimation for the salt-finger double diffusive fluxes is relied on the parameterization for very simple situation. Discussion using the microstructure measurements should be done to evaluate the validity of the parameterization. The estimate of the density ratio is quite important for the parameterization. The procedure (bin length for gradient estimates) should be further described.

In response to comments by reviewer one, we have comprehensively revised this aspect of the methods. We will no longer divide the water column into gradient and mixed parts of the staircases, but simply implement Equations 10 and 11 over all areas of the water column that are susceptible to salt fingering – as identified using the Turner angle. Furthermore, our newly calculated double-diffusive fluxes (as parameterised in Equations 10 and 11) will be discussed in relation to our direct measurements of turbulent diffusivity from microstructure observations in the context of the density ratio.

4. Discussion is necessary on whether the estimated double diffusive mixing explains water property changes during the present 13 days observations.

We will include further discussion of the influence of double diffusive mixing on water property changes during the observational period.

Minor points:

- Figure 4: figure for $\log10(\varepsilon\mu)$ versus log10() is desirable. From Fig.4c, the differences are both plus and minus.

A scatter plot of $\log10(\varepsilon_\mu)$ versus $\log10(\varepsilon_T)$ will be included in the revised paper.

- 195[th] : Fig. 5a ï 5 ?

The incorrect reference to Figure 5a will be corrected.

- 208[th] : (<10[-5] ) ï (>10[-4.5] ) ?

- 217[th] : ï ?

We had neglected to include units after $10^{-1}$ on line 217: this will be corrected.

---

## Author Response (AR1)

**Response to reviewer 1**

Throughout this document, the two reviews are reproduced in black. Our responses are presented in blue and, where appropriate, quotations from the revised paper are included in indented *italic blue* text.

In this manuscript, Sheehan et al. describe results from a Seaglider mission in January-February 2020 in the western tropical Atlantic. The paper focuses on reporting dissipation rates of turbulent kinetic energy using two methods: a direct microstructure method, with which temperature fluctuations are measured at centimetre-scales, and an indirect method based on the meter-scale vertical overturns detected in the temperature profiles (the so-called Thorpe method), also measured with a microstructure probe. The most interesting pattern in the observations is a local enhancement of turbulent dissipation below a subsurface layer of high-salinity associated with the subtropical underwater. The authors claim that this is the first study reporting microstructure temperature measurements from this particular type of glider. Another highlight of the paper is that, contrary to some previous assessments, the direct and indirect methods for estimating dissipation rates agree well in their dataset. Finally, the authors also report and discuss heat and salt fluxes driven by the measured turbulent motions and compare them with those potentially caused by double-diffusive instabilities, for which the conditions are favourable in the region.

Although the paper has some strong points (it describes novel dataset with some interesting results, it is well written), it has also – in my view – some major weaknesses that need to be addressed before publication. My general feeling is that the authors missed several good opportunities to make the paper more relevant and useful for their colleagues. I list my comments below with some suggestions of further analysis/discussions that could potentially improve the impact of the article:

We thank the reviewer for recognising the value of the dataset and the work, and for the thorough suggestions they have made regarding its improvement. We agree that they will increase the impact of the article. The authors feel that this review has been particularly helpful – and must have taken considerable time and effort, for which we are very grateful. The paper has been greatly improved as a consequence of the reviewer's input.

(1) Microstructure Method. Overall, I missed a more extensive and direct acknowledgement of the limitations of the different methods used. The authors use a nowadays standard method to estimate energy dissipation rates from temperature microstructure measurements (the Batchelor fitting method). However, from my experience and the available literature, the method has important limitations and the results may be highly dependent on some user choices. In my opinion, those aspects are relevant to discuss. One key limitation is the fact that the dissipation rates detectable with the method are upper bounded due to the limited time response of the FP07 sensor. For typical glider along-path speeds of 40 cm/s, maximum dissipation rates measurable are about 10 W/Kg. The authors report values larger than that in the upper mixing layer, but I haven't seen any mentioning as to how the sensor limitation may affect their results, in particular those values, and the comparison with the Thorpe-scale method. I would like to see some of the temperature-gradient spectra in the upper mixing layer and the fit parameters, to confirm that those values correspond to real dissipation. My experience is that for high

epsilon values the roll-off in the temperature gradient is probably imposed by sensor limitations, such that the Batchelor fitting is not really meaningful (see also Paterson and Fer 2014).

We thank the reviewer for bringing this limitation to our attention. Following Peterson and Fer (2014), we now remove any estimate of epsilon greater than $2 \times 10^{-7}$ W kg$^{-1}$. We have added this as a quality control criterion in Section 2.2:

> Line 132. "The estimate of $\varepsilon_\mu$ is greater than $2 \times 10^{-7}$ W kg$^{-1}$; even having corrected estimates of $\Lambda_{32}$ at high frequencies, the effective resolution of the fast thermistor (100 Hz) is such that values of $\varepsilon_\mu$ greater than this cannot be reliably estimated."

Furthermore, we have now outlined the expected influence of sensor limitation on our results in Section 3.1:

> Line 202. "But these highest values [$\varepsilon_\mu > 10^{-8}$] are infrequently observed [in the surface mixed layer]: a large proportion of $\varepsilon_\mu$ estimates in the mixed layer are greater than $2 \times 10^{-7}$ W kg$^{-1}$, beyond the range for which the FP07 fast thermistor observations and the Batchelor spectrum method yield meaningful results (Sec. 2.2; Peterson and Fer, 2014). Consequently, the remaining values are all below this threshold; indeed, considerable variability in $\varepsilon_\mu$ is observed in the mixed layer, and many remaining estimates are low ($< 10^{-10}$ W kg$^{-1}$). The mean $\varepsilon_\mu$ in the upper water column is therefore likely to be biased towards these low values."

The time-depth plot of estimates of $\varepsilon_\mu$ with the addition of the new upper-limit cut off ($\varepsilon_\mu > 2 \times 10^{-7}$ W kg$^{-1}$) is presented in Figure 5a of the revised paper. Examples of relatively high $\varepsilon_\mu$ ($10^{-8} < \varepsilon_\mu < 2 \times 10^{-7}$ W kg$^{-1}$) spectra from the mixed layer have been presented in the online Authors' Response (24 October 2022.)

Further, the temperature spectra are typically partially corrected for the sensor time-response. The best parameters for the adjustment are not well stablished and can differ from sensor to sensor. It would be good to report explicitly what exact correction is used here.

We apologise for omitting this information in our summary of the method of Scheifele et al. (2018). Once the temperature power spectrum, $\Lambda_{32}$, has been calculated from a 32-second segment, the transfer function of Sommer et al. (2013) is used to correct for the slow FP07 response time at high frequencies. This step is included in the Matlab toolbox provided by Ben Scheifele and Jeffrey Carpenter. We have added this step into our summary of the method:

> Line 90. "Values of $\Lambda_{32}$ at high frequencies, where the thermal inertial of the fast thermistor is such that its temporal response is inadequate, are corrected using the transfer function of Sommer et al. (2013)."

I also have a general vocabulary concern. The authors make a difference between what they call "spectral" methods for calculating turbulent dissipation, as opposed to non-spectral methods, like the fine-scale and Thorpe methods. In my view this is not entirely correct since, for example, the fine-scale method often involves spectral calculations. In my opinion, what distinguishes the two kind of methods is the scale at which measurements are done. In this regard, I think it would be more correct to use the term "microstructure method" as opposed to "indirect methods" (or something similar). I would recommend to change this throughout the manuscript.

We agree that, on reflection, spectral methods does not accurately and uniquely describe the methods used in this study. In the revised paper, we have used "microstructure" methods/observations to refer to the Batchelor spectrum method applied to 512 Hz FP07 temperature observations; we have used "finescale" methods/observations to refer to observations of lower resolution (e.g. ship and glider CTD) and to the methods commonly used to process them (e.g. Thorpe scale, finescale parameterisation).

> *Line 23. "Methods such as Thorpe scaling (Thorpe, 1977) and finescale parameterisation (Polzin et al., 2014; Whalen et al., 2015) have been developed to enable ε to be estimated from ordinary CTD and ADCP observations of temperature, salinity and velocity – hereafter referred to as finescale methods and observations. Although finescale methods do not require specialist instruments (e.g. Fer et al., 2010b; Whalen et al., 2012, 2015), they are dependent on more assumptions; their results tend not to be valid over as wide a range of conditions as those such as the Batchelor spectrum method (Batchelor, 1959) applied to high-resolution observations (Polzin et al., 2014; Whalen, 2021) – hereafter referred to as microstructure methods and observations."*

(2) Thorpe-scale method. Regarding the Thorpe method, its applicability to non-vertically profiling instruments like gliders has been questioned in the past (e.g. Thorpe et al. 2012), since the method is based on measuring the vertical size of turbulent overturns, which is not entirely achievable with a slant-wise measuring trajectory. Although your results are encouraging, I still think it would be better to acknowledge and discuss the potential impact of this limitation more explicitly.

We thank the reviewer for highlighting this potential limitation of using the Thorpe scale method on glider observations. Thorpe (2012) suggested that internal waves, when present, can lead to false overturn detection and therefore to artificially high estimates of $\varepsilon_T$; Howatt et al. (2021) further suggest that the detection of false overturns is of concern when the slope of internal waves is greater than the glide slope (i.e. angle from the horizontal) of the glider. Following Howatt et al. (2021), we calculate that the mean (± one standard deviation) glide slope for the present mission is 35 ± 5 degrees – i.e. greater than the maximum slope of internal waves (16 degrees). We have acknowledged this in the revised paper:

> *Line 326. "A potential problem with using the Thorpe scale method on Seaglider observations is the non-vertical profile that the Seaglider collects: unlike, for instance, a ship-deployed CTD or a vertical microstructure profiler, a Seaglider follows a slanted trajectory, typically covering a horizontal distance of approximately 4 km over a 1000 m-deep dive-climb cycle. Any resultant sampling of horizontal gradients, particularly in the presence of internal waves, could artificially inflate estimates of $\varepsilon_T$ due to false overturn detection (Thorpe 2012). However, this is a concern only when the internal wave slope exceeds the slope of the Seaglider's trajectory. Following Howatt et al. (2021), we calculate that the mean (± one standard deviation) of SG620's trajectory slope (from the horizontal) during the Eurec4a deployment was 35 ± 5°, i.e. greater than the upper limit of the slope of internal waves (16°; Thorpe, 1978). Consequently, false overturn detection due to the Seaglider's sloping trajectory is unlikely to lead to over-estimation of $\varepsilon_T$ in this dataset."*

Also, the authors contrast their results (in which the microstructure and Thorpe methods agree well), with a previous study (Howatt et al. 2021) in which the Thorpe method tended to overestimate dissipation. They speculate that this difference may be due to the different probe resolution: in the present study a microstructure probe is used, which allows a much better resolution of the smaller overturns, whereas Howatt et al. (2021) use a standard CTD probe. While this is just a mere speculation, the authors have the available data to evaluate this hypothesis: they could apply the Thorpe method to the glider regular CTD data and compare the results with those of the FP07-Thorpe method. I think not

doing that is a missed opportunity, since it would be very useful for future studies to know whether the disagreement comes from the different spatial resolution or not.

We thank the reviewer for this suggestion. We feel that the original wording in our discussions did not fully convey the weight of the findings of Howatt et al. (2021), who clearly conclude that the low resolution of a glider CTD sets a minimum resolvable turbulent patch size. In oceanographic conditions where overturns are likely to occur over much smaller scales than the minimum resolvable turbulent patch size, ensuing estimates of $\varepsilon_T$ will necessarily be biased high. We agree with this argument. Our original discussion should have made this clearer. Values of $L_T$ derived from the FP07 fast thermistor observations (512 Hz) are clearly far smaller than those derived from ordinary Seaglider observations (0.2 Hz; see Figure 3 of the Authors' Response, 24 October 2022), indicating that the latter are unable to resolve the majority of overturns in this region. Estimates of $\varepsilon_T$ from these ordinary Seaglider observations will therefore be extremely unreliable. Given that Howatt et al. (2021) have already reported this conclusion, we do not want to repeat their work and include figures illustrating this difference, but we have commented on the inability of the ordinary Seaglider observations to accurately resolve $L_T$:

> *Line 309. Our finding that estimates of $\varepsilon_T$ are higher than estimates of 75 $\varepsilon_\mu$ is in agreement with the findings of Howatt et al. (2021). They report that, while the Thorpe scale method can well rep- resent qualitative aspects of the spatio-temporal distribution of $\varepsilon$, the method can, when applied to relatively low resolution finescale observations, artificially inflate $\varepsilon_T$ by an order of magnitude relative to estimates from, for instance, the Batchelor spectrum method. This is for the simple reason that finescale observations lack the resolution to capture small-scale overturns; the resultant distribution of LT, and hence of $\varepsilon$ is consequently biased towards high values.*
>
> *"This is supported by our observations: the differences between $\varepsilon_\mu$ and $\varepsilon_T$ reported here are much lower, being, on average, far less than an order of magnitude (Sec. 3; Figs. 5 and 6). Further, we directly compare $L_T$ calculated using the FP07 microstructure observations (512 Hz, 100 Hz when accounting for the sensor's response time; Fig. 3) to $L_T$ calculated using the Seaglider's ordinary finescale temperature and salinity observations (0.2 Hz). The distribution of LT derived from microstructure observations has a pronounced peak at very low values (> 0.25 m; Fig. 10). In contrast, the finescale glider observations are unable to resolve values of $L_T$ at this scale (Fig. 10). The better agreement between $\varepsilon_\mu$ and $\varepsilon_T$ reported here than by Howatt et al. (2021) therefore appears to be explained by the higher vertical resolution of our observations. Consequently, and in agreement with Howatt et al. (2021), we suggest that the Thorpe scale method is unable to produce reliable quantitative estimates of $\varepsilon$ unless it is applied to microstructure observations. In a high-turbulence environment where overturns are large, finescale observations may be better able to accurately resolve the distribution of $L_T$ and their relatively low resolution may not introduce such a systematic bias."*

I have another concern about the method, in particular about what you describe in line 145 regarding the aggregation of overturns until they reach a scale of 2 m. Sorry, I am not familiar with this approach, but it seems a bit weird to me, why do you need to impose a minimal overturning scale if those appear smaller in the data? This could bias high the lower dissipation rates, couldn't it?

The aggregation of overturns is an established method, as used by both Ijichi and Hibiya (2018) and Howatt et al. (2021). The former study uses 512 Hz VMP observations and aggregates overturn patches until they reach a minimum heigh of 5 db; the latter uses 0.5 Hz Slocum glider observations and aggregates overturn patches until they reach a minimum height of 2 m. Larger overturns are typically indicative of recently formed turbulent patches starting the dissipation process; conversely, a number of small overturns observed over a short section of a profile is likely to indicate a previously larger

overturn at the tail-end of the dissipation process (Smyth et al., 2001). To account for this, and to be consistent with previous work, we aggregate overturns, but using the more conservative of the above published values (i.e. 2 m). We emphasise that aggregation only occurs when consecutive overturns are separated by less than 1 m; overturns less than 2 m in height that are more than 1 m from the preceding or following overturns are not aggregated.

(3) Salt fingers. I have two main comments regarding the double diffusion diffusivity calculations and fluxes. First, I think there is a clear missed opportunity here of using the microstructure data to assess the role of double diffusion more directly. I missed, for example, a figure showing some of the staircases, but more importantly, you could use the thermal variance dissipation rates to directly assess the rate of heat mixing in the salt-finger favourable zone using the Osborn-Cox method. Several authors (e.g. St Laurent et al. 1999; Schmitt et al. 2005) have reported levels of thermal variance dissipation in thermohaline staircases, and heat diffusivities above those predicted by the Osborn model (your eq. 7). In most microstructure studies based on shear probes this information is missing, but you have it and you could compare the results with the flux law estimates.

As discussed immediately below, we have changed the method used to identify those regions susceptible to salt fingering. Furthermore, we have added a figure illustrating the staircase structures (Figure 8 of the revised paper). We agree that an estimate of heat diffusivity based on our estimates of $\chi$ makes for an interesting comparison with the existing results and have included such an estimate in the revised paper (Figure 9c and d). This figure is discussed in Section 3.3:

*Line 270. We use the Osborn and Cox (1972) relation to estimate $\kappa_\Theta$ and $\kappa_S$ using $\chi$ (see also e.g. St Laurent and Schmitt, 1999; Schmitt et al., 2005; Ijichi and Hibiya, 2018), which we calculated as an intermediary step in the calculation of $\varepsilon_\mu$ (Sec. 2.2):*

*$$\kappa_\Theta = \chi / 2\Theta_z^2$$*

*The relationship between $\kappa_\Theta$ and $\kappa_S$ when calculated from $\chi$ remains as in Eqn. 11 (Schmitt et al., 2005; van der Boog et al., 2021). We refer to these second salt finger diffusivities as Osborn-Cox $\kappa_\Theta$ and $\kappa_S$.*

Relevant excerpts from the paper are too lengthy to include here, but we draw the reviewer's and editor's attention to Section 3.3, which has been substantially re-written to discuss these results as suggested by the reviewer.

My second comment concerns the latter method. I see that in your Figure 7, salt fingers diffusivities are zero except in the step layers, however it is not clear how you deal with those zeros in the calculation of mean diffusivities and fluxes. If you just neglect them, you may be overestimating the average diffusivities by quite a lot! But this is not explicit. Can you clarify that? That said, I am not entirely sure about the way in which the flux-law formulae should be applied, but my intuitive guess would be that the equations should be applied for the entire region of the vertical profile where the density ratio is favourable for salt fingers and the steps are observed, that is, applied to the background temperature-salinity gradients instead of the gradients in the steps, as you did. Not sure if I am wrong on that, but I invite you to dive a deeper into it and find out what is the most correct way of applying the formulae and report the details (how and where the diffusivities, gradients and fluxes were calculated; how did you deal with zeros.

Having more closely studied the original paper by Radko and Smith (2012), we agree with the reviewer that Equations 10 and 11 should be applied to the entire region of the vertical profile where the density

ratio is favourable for salt fingers, rather than just in the gradient layers. In the revised paper, we have used the Turner angle, Tu, to identify those parts of the water column that are susceptible to salt fingering, and have applied the equations to these regions. The new diffusivity estimates are presented in Figure 8 of the revised paper. Further, we agree that it is more accurate to calculate a time-depth average while including regions in which the fluxes of heat and salt due to salt fingering are zero – i.e. to include these zeros in our averages.

> *Line 257. Those regions of the water column that are susceptible to salt fingers may be identified using the Turner angle, Tu: salt fingers can occur where $45° < Tu < 90°$. The following equations are applied only where this condition is met. Tu is calculated from temperature and salinity binned into 5 m vertical bins; all subsequent diffusivities are calculated from variables binned into 5 m vertical bins in order to match Tu.*

> *Line 290. Note that, for the calculation of the arithmetic mean heat and salt fluxes in the salt finger regime, we set both $\kappa_T$ and $\kappa_S$ from $R_\rho$ to zero outside of the salt finger regimes, i.e. where $Tu < 45°$ and $Tu > 90°$, and include these zeros in our averages. Because we first filter out regions of the water column that are not susceptible to salt fingering, gaps in the record therefore indicate an absence of the process to be averaged (i.e. salt finger-driven fluxes) rather than an absence of data.*

(4) Discussion of the observed patterns. The main feature of your dissipation dataset is an enhancement in a thick layer below the salinity maximum. In my view, there is also a missed opportunity here of making this result more meaningful and relevant by discussing the possible drivers of this enhancement. Is that maybe due to internal wave interactions with the rapidly changing stratification? Do you have current measurements from the ship that indicate a local enhancement of shear or other data that could give some clue of what is going on? Not sure if further analysis on this is needed for the manuscript (which has a strong methodological component), but I think it could help to make the paper more relevant and interesting.

We agree that discussion of the observed distribution of ε would make for an interesting addition to the paper. However, we regret that we were unable to find processed, quality-controlled datasets that would readily enable us to do so. Furthermore, we note that ships participating in the Eurec4a campaign did not spend much time in the vicinity of the Seaglider.

SPECIFIC COMMENTS:

Line 8. Superscript missing in W kg⁻¹

This has been corrected.

Line 21. The Batchelor method applies only to scalars (e.g. temperature), not shear. The use in this sentence is not correct

This sentence has been modified:

> *Line 21. Estimating the dissipation rate of turbulent kinetic energy, ε, from high-resolution observations of shear and temperature (e.g. Lueck et al., 2002; Peterson and Fer, 2014; Scheifele et al., 2018) has, historically, required considerable ship time, plus specialist instruments and expertise."*

Line 25. These authors report 266 profiles in 50 different stations

We have removed this statement.

Line 61, the use of "," and "()" in this sentence, and other locations in the manuscript, seems weird to me

We have made changes to improve punctuation in various locations in the manuscript.

Line 73. Why didn't you use the shear probe data in this paper? It feels also like a potential missed opportunity of a more thorough evaluation of the different methods.

There was a fault with the shear probe and the data are unfortunately unusable. This has been mentioned in the revised paper:

Line 73. "There was a fault with the shear probe on this deployment and the observations could not be used."

Line 95. This sentence "the rate of destruction of temperature-gradient variance" is not correct: note that chi units are K^2 per second, not (K/m)^2 per second. Temperature of thermal variance is destroyed, not the temperature-gradient variance. Remove "-gradient"

This has been corrected as suggested.

Line 114 – 115. I am not familiar with this paper but the normalisation of the temperature gradient spectra with epsilon seems a bit unphysical to me, since the spectrum scales with chi not with epsilon. There is a proper way of normalising the temperature variance spectrum (see e.g. Dillon and Caldwell (1980)).

The full procedure developed by Scott et al. (2021) is described in the supplementary material to the paper we cited (available at *agupubs.onlinelibrary.wiley.com/doi/full/10.1029/2020JC016861*). We normalise estimates of ε only to perform this quality control step: it is the original, non-normalised values of ε that are subsequently considered in the paper.

Line 123. You could perhaps define the meaning of the term (epsilon/N)^1/2. Also note the wrong spelling in "Tayor"

This sentence has been modified:

Line 125. "If the quantity $U/(\varepsilon_\mu/N)^{1/2}$ is less than five, where U is the glider's speed-in-direction-of-travel and $(\varepsilon_\mu/N)^{1/2}$ is an estimate of the turbulent flow velocities (Fer et al., 2014), indicating that Taylor's frozen turbulence hypothesis is invalid."

Line 161 onwards. How was the critical value chosen? It seems a bit arbitrary. Why don't you use the alpha and beta coefficients to directly estimate the contributions of T and S gradients/variability to density stratification/variability? A more detailed description of the results of this quality check would be desirable. How many datapoints have been discarded? Where?

We thank the reviewer for bringing this to our attention. We agree that using alpha and beta coefficients (i.e. the stability ratio) is more appropriate as a quality control measure. In the revised paper, we have

implemented a quality control check based on the density ratio, and have described the impact of this on our dataset.

> Line 165. "Our estimates of $\varepsilon_T$ are derived from potential temperature rather than potential density; we must therefore assume that temperature is the dominant control on density. In regions where this is not the case – i.e. in regions where salinity is the dominant control on density – temperature perturbations may not correspond to the density perturbations that the Thorpe scale method takes to be indicative of turbulent overturns. To identify regions where salinity is the dominant control on density, we use the density ratio, $R_\rho$:
>
> $$R\rho = \alpha\Theta_z/\beta S_z$$
>
> where $\alpha$ is the thermal expansion co-efficient, $\Theta_z$ is the vertical temperature gradient, $\beta$ is the haline contraction coefficient and $S_z$ is the vertical salinity gradient. We calculate $\Theta_z$ and $S_z$ from finescale glider observations binned into 5 m vertical bins using the arithmetic mean, and find $R_\rho$ at the depth of each overturn. Where $-1 < R_\rho < 1$, salinity is the dominant control on density, and we discard any overturns and associated value of $\varepsilon_T$. In total, 2388 overturns are discarded by the $R_\rho$ quality control criterion, 4.03% of the total. In the majority of the water column, temperature is the dominant control on density, and so temperature observations may be reliably used to estimate $\varepsilon$. Estimates of $\varepsilon_T$ that are discarded correspond principally to large values of $L_T$ (> 1 m) in mid-depth regions (i.e. between 200 and 600 m). This is the part of the water column in which the majority of the thermohaline staircases are found (Rollo et al., 2022)"

Figure 3 or 4. It could be nice to see a distribution of LT here or elsewhere.

We agree that this makes a nice addition to the paper. A time-depth section of $L_T$ is presented in Figure 3 of the revised paper.

Figure 4. Why are there no epsilon_T values in the upper mixing layer? What is the criterion to discard those values? (I guess it is related to the very weak stratification but you should mention it)

Values of $\varepsilon_T$ were removed above 75 m, as there is an apparent large overturn in temperature at approximately 60 m depth. From the glider CTD data, it is clear that this is density-compensated and is therefore not a turbulent overturn. This has been added to the text:

> Line 179. "Finally, we discard all values of $\varepsilon_T$ shallower than 75 m because a temperature inversion in the mixed layer is erroneously identified as an overturn. Remaining estimates of $\varepsilon_T$ are binned into 25 m vertical bins using the geometric mean, as for $\varepsilon_\mu$."

Section 3.1. Uncertainties reported in this section have inconsistent values and units, also with respect to Figure 4. E.g. values in line 180, are much higher than those in 184 (and have no units, while those in 184 do have units), and they also seem different from the error margins in the figure. Please, clarify how the standard deviations are calculated and improve the consistency.

We thank the reviewer for pointing out these inconsistencies, in particular the incorrect use of units when discussing the dimensionless geometric standard deviation factor in line 184. We have not included a definition of the geometric standard deviation factor in the manuscript as this can be found in standard mathematical textbooks. However, given that many readers will not be familiar with geometric means and geometric standard deviation factors, we will expand the descriptions given here to provide some clarification:

*Line 135. "Quality-controlled estimates of $\varepsilon_\mu$ were binned, profile by profile, into 25 m bins; we use the geometric mean in preference to the arithmetic mean, the better to represent the average of observations that span many orders of magnitude and which are not normally distributed."*

*Line 195. "The more variable of the two is $\varepsilon_\mu$, the geometric standard deviation factor (GSDF) of which is 5.58. The GSDF of $\varepsilon_T$ is 4.43. (Note that GSDF is multiplicative, not additive, and is therefore dimensionless. The range is from the geometric mean/GSDF to geometric mean × GSDF.) Averages and GSDFs are calculated from bins only where estimates of both $\varepsilon_\mu$ and $\varepsilon_T$ are available."*

Throughout the paper, we have clarified when we are using the arithmetic or geometric mean and standard deviation.

Line 195. There is no panel a in Figure 5.

This has been corrected.

Lines 195 – 196. Not sure I see this modest peak you talk about, or maybe it is too modest to be worth mentioning?

This sentence has been removed.

Line 199 – 200. Is there something wrong with this sentence? I do not get it.

The wrong number was used to illustrate this point. This has been corrected:

*Line 216. "This is in contrast to higher values of $\varepsilon_\mu$ and $\varepsilon_T$ (> $10^{-10}$ W kg$^{-1}$) within the 700 to 800 m depth range at the beginning and end of the deployment."*

Line 205. Why? (see comment above)

As mentioned above, values of $\varepsilon_T$ were removed above 75 m, as there is an apparent large overturn in temperature at approximately 60 m depth.

Line 206 – 207. I think you need a bit more clarity in this sentence. What makes the epsilon and kappa distributions similar?

This sentence has been changed:

*Line 223. "Below approximately 250 m, the distribution of $\kappa_\rho$ resembles that of $\varepsilon_\mu$ due to the relatively low variability in $N^2$ at these depths (Fig. 3c). Above 200 m $N^2$ increases substantially in the pycnocline and thus $\kappa_\rho$ decreases."*

Line 216. Epsilon units missing

Units have been added.

Line 217. Is there an upward diffusive heat flux in the upper layer? Does that mean that thermal stratification is inverse?

There is indeed a small thermal inversion near the surface – i.e. the temperature maximum (> 28°C) is found at a depth of approximately 60 m. Above this, temperature decreases by approximately 0.5°C. This is not visible in Figure 4a given the contour interval. We have explained this in the revised paper in relation to the upward heat flux visible in Figure 7b and have added lines to Figure 7a and b indicating, respectively, the depths of the temperature and salinity maxima.

*Line 235. "Within the surface mixed layer, notwithstanding the limited coverage of the observations, $Q_h$ is positive in the top 50 m and negative between 50 and 100 m; by contrast, $Q_S$ is positive throughout the surface mixed layer (Fig. 7b and c). This is due to weak thermal and haline inversions near the surface; the depths of the temperature and salinity maxima are indicated by the black lines in Fig. 7b and c respectively."*

Line 235. "[...] and given that theoretical flux laws can overestimate κS in the real ocean [...]". This sentence needs a reference.

References to Taylor and Veronis (1996), Kelley et al. (2003) and Radko (2005) have been added in the revised paper.

Line 245. By eye this difference looks greater. More details on how the averaging is performed are needed.

We have refined the method by which we selected those parts of the water column in which we calculate the fluxes of heat and salt due to salt fingering (see response to main comment 3, page 5 above). We have provided more details on how the averages are performed – in particular including zeros in between salt fingering regions.

Line 258. "sampling and the necessary setting of a minimum Thorpe scale", why do you need a minimum Thorpe scale? (see comment above)

We apologise, the reference to setting a minimum Thorpe length scale was an error: the original text should have referred to the setting of a minimum length scale for a turbulent patch. No minimum Thorpe scale was or has been set. The discussion of the influence of resolution on the reliability of $\varepsilon_T$ estimates has been thoroughly revised (line 326) but the original, erroneous sentence has been removed. Our implementation of the Thorpe scale method is described in Section 2.3.

Lines 262 – 263. "however, the lack of similarly high-resolution salinity observations means that the resolution of N used to calculate εT in Eqn. 5 will limit any improvement that a small minimum LT might otherwise have on estimate of εT". The N that you introduce in the εT formula is, in my view, a background N representative of scales larger than the turbulent overturns, so I don't necessarily see the lower resolution of N as a problem here.

We agree with the reviewer that the N that we introduce in the $\varepsilon_T$ formula is meant to be a background N representative of scales larger than the turbulent overturns. Therefore, the lack of high-resolution salinity observations is not a problem. We have removed this sentence from the revised paper and have calculated N from temperature and salinity observations averaged into 5 m bins, the smoothed over 45 in the vertical (Gaussian window running arithmetic mean).

*Line 162. "... where N is the background buoyancy frequency calculated using the Seaglider's finescale temperature and salinity observations, binned into 5 m vertical bins, then smoothed in the vertical using a Gaussian-windowed running mean over nine bins (i.e. 45 m)".*

Lines 264—265. "Howatt et al. (2021), who reduce the difference between their two estimates of ε," I don't get the meaning of this sentence

This sentence has been removed.

Line 285. See main comment (4)

Please see our response to main comment (4)

Lines 290 – 292. "The downward flux of heat due to double diffusion is almost four times that due to turbulent mixing; the downward flux of salt due to double diffusion is 7.5 times that due to turbulent mixing." The differences in diffusivities (as reported in line 245), appear much weaker. More details are needed on this calculation.

The depth-time distributions of thermal ($K_\Theta$) and haline ($K_S$) diffusivity do not match the depth-time distribution of turbulent diffusivity ($K_\rho$). One can imagine, therefore, a point with a large temperature gradient, $\Theta_z$, a large $K_\rho$ but a small $K_\Theta$; and another point with a small $\Theta_z$, a small $K_\rho$ but a large $K_\Theta$. The ratio of average $K_\rho.\Theta_z$ to average $K_\Theta.\Theta_z$ will not be the same as the ratio of $K_\rho$ to $K_\Theta$.
        We note that the diffusivity and flux calculations have been thoroughly revised in line with the reviewer's suggestions – including the method used to identify those regions of the water column susceptible to salt fingering. Average diffusivities in the salt finger regimes with zeros included (see comment above on page 6) are not quoted, as the geometric mean of a dataset that contains a zero is always zero (in contrast to the arithmetic mean). We have quoted arithmetic means of the resultant fluxes, which may be calculated using the arithmetic mean: we draw attention to Section 4.3, and in particular to the paragraphs starting on line 284 and line 299. We note that the differences between average fluxes in the turbulent and salt finger regimes are now less than previously reported, and appear to be a more accurate reflection of the differences in the underlying diffusivities (Figures 7 and 9)

Lines 298 – 300. "We posit that this improved agreement is due to εT being calculated using the same high-resolution observations as were used to calculate εμ: other studies have compared εμ to εT calculated using lower-resolution temperature observations." That seems plausible but you have the data to test this and I think you should (see main comment (2)

Please see our response to main comment (2).

**Response to reviewer 2**

Throughout this document, the two reviews are reproduced in black. Our responses are presented in blue and, where appropriate, quotations from the revised paper are included in indented *italic blue* text.

Overall:
This paper described turbulent energy dissipation rate obtained by glider observations from two methods: one is estimated by estimating Batchelor wavenumber using fast-response thermistor and the other is from Thorpe-scale density overturn. The former is less than the latter one. The turbulent heat and salt fluxes are reported not to be influential, while the salt-finger double diffusive fluxes could be influential. These results are potentially worth to be published, but need to be further analyzed, because both the discussions why the difference between the two methods and for the salt-finger fluxes are not enough.

We thank the reviewer for their feedback and suggestions for improvement, which we think have greatly enhanced the paper. We have expanded and strengthened the analysis and discussion as following the suggestions of both reviewers.

Specific points:
1. Methods of estimating the is not enough. Time response of the fast-response thermistor probe is not enough and needs to be corrected even for the glider observations. The underestimation from the spectrum fit could be caused by insufficient spectrum correction. Also the original method for estimating Batchelor wavenumber was proposed by Oakey (1982) and Ruddick et al. (2000) which should be cited.

We apologise for omitting information on the response time correction in our summary of the method of Scheifele et al. (2018). Once the temperature power spectrum, $\Lambda_{32}$, has been calculated from a 32-second segment, the transfer function of Sommer et al. (2013) is used to correct for the slow FP07 response time at high frequencies. This step is included in the Matlab toolbox provided by Ben Scheifele and Jeffrey Carpenter. We have added this step into our summary of the method.

> Line 90. *"Values of $\Lambda_{32}$ at high frequencies, where the thermal inertial of the fast thermistor is such that its temporal response is inadequate, are corrected using the transfer function of Sommer et al. (2013)."*

We have added the suggested references to the revised paper:

> Line 94. *"We transform each $\Delta_{32}$ into a temperature-gradient spectrum, $\Psi$, which should resemble the Batchelor spectrum, $\Psi_B$ (Batchelor, 1959), the theoretical spectrum that describes temperature-gradient spectra and which is commonly used when calculating $\varepsilon_\mu$ (e.g. Oakey, 1982; Ruddick et al., 2000; Peterson and Fer, 2014; Scheifele et al., 2018)."*

2. The regression coefficient between the Thorpe and Ozmidov scales are also not decisive, and the Thorpe-scale method cannot estimate for depths without density inversions. The standard method for

estimating is shear-probe measurements. Before comparing between the Batchelor and Thorpe methods, comparisons between shear and Batchelor and between shear and Thorpe methods should be discussed or appropriate previous studies should be cited.

While the regression coefficient between the Thorpe and Ozmidov scales ($R_{OT}$) is indeed not constant, the work of Ijichi and Hibya (2021) demonstrates that $R_{OT}$ is not highly variable in the upper 1000 m of the water column. We feel that a full treatment of this matter is outside the scope of the present paper. Furthermore, there was a fault with the shear probe and the data are unfortunately unusable. This has been mentioned in the revised paper:

> Line 73. "There was a fault with the shear probe on this deployment and the observations could not be used."

3. The estimation for the salt-finger double diffusive fluxes is relied on the parameterization for very simple situation. Discussion using the microstructure measurements should be done to evaluate the validity of the parameterization. The estimate of the density ratio is quite important for the parameterization. The procedure (bin length for gradient estimates) should be further described.

In response to comments by reviewer one, we have comprehensively revised this aspect of the methods. We no longer divide the water column into gradient and mixed parts of the staircases, but simply implement Equations 10 and 11 over all areas of the water column that are susceptible to salt fingering – as identified using the Turner angle. Furthermore, our newly calculated double-diffusive fluxes (as parameterised in Equations 10 and 11) are discussed in relation to our direct measurements of turbulent diffusivity from microstructure observations in the context of the density ratio.

Relevant excerpts from the paper are too lengthy to include here, but we draw the reviewer's and editor's attention to Section 3.3, which has been substantially re-written.

4. Discussion is necessary on whether the estimated double diffusive mixing explains water property changes during the present 13 days observations.

The average turbulent and double diffusive fluxes over the period of the observations has been presented. We have also estimated the change in temperature and salinity of the Subtropical Underwater layer that these fluxes would drive over a year – assuming the estimated fluxes are representative of annual-mean conditions.

> Line 240. "Over the period of the observations, the arithmetic mean $Q_h$ between 200 and 500 m was −1.40 W m$^{-2}$. This, and all subsequent flux estimates, are summarised in Table 1. Over the period of the observations, the arithmetic mean $Q_S$ between 200 and 500 m was −5.84 × 10$^{-8}$ kg m$^{-2}$ s$^{-1}$. This is a relatively low-turbulence region; the attendant turbulent fluxes are correspondingly relatively small and likely have little influence on the region's hydrography. For instance, integrated over a year, $Q_h$ results in an annual turbulent heat flux of 15 −4.43 × 10$^7$ J m$^{-2}$, which would reduce the temperature of the overlying SUW layer (assumed to be 100 m thick) by just 0.11°C. Similarly integrated over a year, $Q_S$ results in an annual turbulent salt flux of −1.84 kg m$^{-2}$, which would reduce the salinity of the overlying SUW layer by just 0.02 g kg$^{-1}$."

> Line 284. We then substitute $\kappa_\Theta$ from $R_\rho$ (Fig. 9a) into Eqn. 8 in place of $\kappa_\rho$ (Fig. 7a). Averaged (arithmetic mean) between 200 and 500 m, $\kappa_\Theta$ from $R_\rho$ gives rise to a heat flux of −1.71 W m$^{-2}$ in the salt-finger regime (Table 1), an annual temperature reduction of 0.13°C in the SUW layer. This flux is larger than the value reported above for the turbulent regime (Sec. 3.2). Similarly, we substitute $\kappa_S$ from $R_\rho$ (Fig. 9b) into Eqn. 9 in place of $\kappa_\rho$. Averaged (arithmetic mean) between 200

*and 500 m, $\kappa_S$ from $R_\rho$ gives rise to a salt flux of −1.83 × 10⁻⁷ kg m⁻² s⁻¹ in the salt-finger regime (Table 1), or an annual reduction of 0.06 g kg⁻¹ in the salinity of the SUQ layer. This flux is over three times larger than the corresponding salt flux in the turbulent regime.*

*Line 299. We then substitute $\kappa_\Theta$ from the Osborn-Cox relation (Fig. 9c) into Eqn. 8 in place of $\kappa_\rho$. Averaged (arithmetic mean) between 200 and 500 m, $\kappa_\Theta$ from the Osborn-Cox relation gives rise to a heat flux of −1.49 W m⁻² in the salt-finger regime (Table 1), an annual temperature reduction of 0.11°C in the SUW layer. This heat flux is very similar to that reported for the turbulent regime, and less than that predicted by the empirical relation of Radko and Smith (2012) reported above (i.e. $\kappa_\Theta$ and $\kappa_S$ from $R_\rho$). And $\kappa_S$, averaged (arithmetic mean) between 200 and 500 m, gives rise to a salt flux of −9.40 × 10⁻⁸ kg m⁻² s⁻¹ (Table 1), an annual salinity reduction of 0.03 g kg⁻¹ in the SUW layer. This flux is approximately 1.6 times that reported for the turbulent regime, but half that predicted by the empirical relation of Radko and Smith (2012).*

Minor points:

- Figure 4: figure for $\log 10(\varepsilon\mu)$ versus $\log 10()$ is desirable. From Fig.4c, the differences are both plus and minus.

A scatter plot of $\log 10(\varepsilon_\mu)$ versus $\log 10(\varepsilon_T)$ has been included in the revised paper (Figure 6b).

- 195[th]: Fig. 5a ï 5 ?

The incorrect reference to Figure 5a has been corrected.

- 208[th]: (<10⁻⁵) ï (>10⁻⁴·⁵ ) ?

- 217[th]: ï   ?

We had neglected to include units after 10⁻¹ on line 217: this has been corrected.

---

## Author Response (AR2)

**Response to reviewers**

We thank the reviewers for their positive feedback on our revised draft. We have been happy to make the technical corrections requested. We have also included a new DOI for the $\varepsilon_\mu$ and $\varepsilon_\mu$ estimates archived at BODC, given the methodological changes made is response to the first round of reviews. This is included in the updated Data Availability section (line 395).

Throughout this document, reviewers' comments are reproduced in black. Our responses are presented in blue and, where appropriate, quotations from the revised paper are included in indented *italic blue* text.

**Line 106** Inertial -> inertia

This mistake has been corrected.

**Line 291** $10^{-5}$ W/kg seems too large, is this a typo?

The figure quoted ($> 10^{-5}$ W kg$^{-1}$) is correct, but we erroneously quoted this as being a representative value of $\varepsilon$. In fact, the figure is a representative value of $\kappa_\rho$, hence the following reference to Figure 7a. This mistake has been corrected.

**Section 3.3** The Osborn and Cox diffusivities can include the contributions of mechanical turbulence to chi. Perhaps it would be good to acknowledge this.

*Line 275. We refer to these second salt finger diffusivities as Osborn-Cox $\kappa_\Theta$ and $\kappa_S$; we note that the Osborn-Cox relation can include a contribution of mechanical mixing on $\chi$ and hence on $\kappa_\Theta$.*

**Lines 446–447** Although it is true that epsilon and chi are derived from the same data, they can sometimes have very different vertical patterns (driven to a large extent by stratification). So, I do not fully agree with this sentence.

We have removed this and the following sentence from the paragraph, which now reads:

*Line 354. The distribution of $\kappa_\Theta$ and $\kappa_S$ from the Osborn-Cox relation (i.e. from $\chi$), resembles that of $\kappa_\rho$ – itself derived from $\varepsilon_\mu$ – far more closely than do either $\kappa_\Theta$ and $\kappa_S$ from $R_\rho$ (Figs. 7a and 9). Notably, neither $\kappa_\Theta$ nor $\kappa_S$ from $R_\rho$ seems to be particularly influenced by features of the water column that might be expected to influence vertical diffusivity, such as stratification (Fig. 4c) or temperature and salinity gradients (not shown). Given that the Osborn-Cox relation explicitly relates diffusivity to a mixing variable (i.e. $\chi$), we suspect that it is more accurate than the empirical relation of Radko and Smith (2012, i.e. $\kappa_\Theta$ and $\kappa_S$ from $R_\rho$).*

---

## Author Response (AR3)

**Response to editor**

L.235, Please change the unit in the bracket (currently for dissipation rate) to that for diffusivity.

We apologise for this oversight. This mistake has been corrected.